# VideoFactory: Swap Attention in Spatiotemporal Diffusions for Text-to-Video Generation

## Abstract

We present *VideoFactory*, an innovative framework for generating high-quality open-domain videos. *VideoFactory* excels in producing high-definition (1376×768), widescreen (16:9) videos without watermarks, creating an engaging user experience. Generating videos guided by text instructions poses significant challenges, such as modeling the complex relationship between space and time, and the lack of large-scale text-video paired data. Previous approaches extend pretrained text-to-image generation models by adding temporal 1D convolution/attention modules for video generation. However, these approaches overlook the importance of jointly modeling space and time, inevitably leading to temporal distortions and misalignment between texts and videos. In this paper, we propose a novel approach that strengthens the interaction between spatial and temporal perceptions. In particular, we utilize a swapped cross-attention mechanism in 3D windows that alternates the "query" role between spatial and temporal blocks, enabling mutual reinforcement for each other. To fully unlock model capabilities for high-quality video generation, we curate a large-scale video dataset called HD-VG-130M. This dataset comprises 130 million text-video pairs from the open-domain, ensuring high-definition, widescreen and watermark-free characters. Objective metrics and user studies demonstrate the superiority of our approach in terms of per-frame quality, temporal correlation, and text-video alignment, with clear margins.

## 1 Introduction

Automated video production is experiencing a surge in demand across various industries, including media, gaming, film, and television (Joshi et al., 2017; Menapace et al., 2021). This increased demand has propelled video generation research to the forefront of deep generative modeling, leading to rapid advancements in the field (Ho et al., 2022b; Mathieu et al., 2016; Saito et al., 2017; Tulyakov et al., 2018; Vondrick et al., 2016). In recent years, diffusion models (Ho et al., 2020) have demonstrated remarkable success in generating visually appealing images in open-domains (Rombach et al., 2022; Ramesh et al., 2022b; Podell et al., 2023). Building upon such success, in this paper, we take one step further and aim to extend their capabilities to high-quality text-to-video generation.

As is widely known, the development of open-domain text-to-video models poses grand challenges, due to the limited availability of large-scale text-video paired data and the complexity of constructing space-time models from scratch. To solve the challenges, current approaches are primarily built on pretrained image generation models. These approaches typically adopt space-time separable architectures, where spatial operations are inherited from the image generation model (Ho et al., 2022b; Hong et al., 2022). To further incorporate temporal modeling, various strategies have been employed, including pseudo-3D modules (Singer et al., 2022; Zhou et al., 2022), serial 2D and 1D blocks (Blattmann et al., 2023; Ho et al., 2022a), and parameter-free techniques like temporal shift (An et al., 2023) or tailored spatiotemporal attention (Wu et al., 2023; Khachatryan et al., 2023b). However, these approaches overlook the crucial interplay between time and space for visually engaging text-to-video generation. On one hand, parameter-free approaches rely on manually designed rules that fail to capture the intrinsic nature of videos and often lead to the generation of unnatural motions. On the other hand, learnable 2D+1D modules and blocks primarily focus on temporal modeling, either directly feeding temporal features to spatial features, or combining them through simplistic element-wise additions. This limited interactivity usually results in temporal

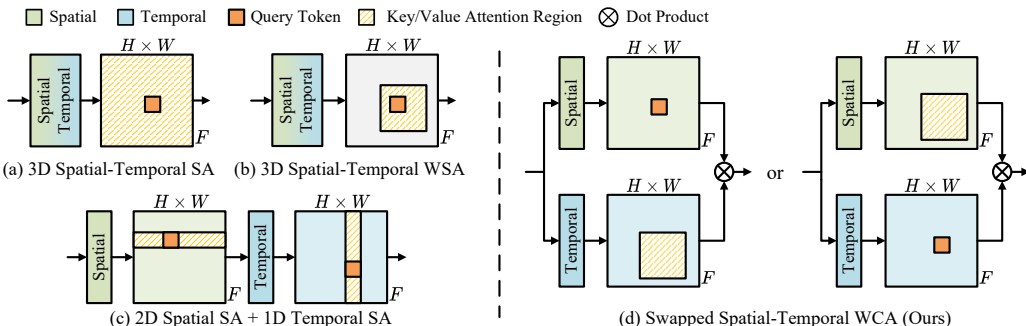

Figure 1: The paradigm of Swapped spatiotemporal Cross-Attention (Swap-CA) in comparison with existing video attention schemes. Instead of only conducting self-attention in (a)-(c), we perform cross-attention between spatial and temporal modules in a U-Net, which encourages more spatiotemporal mutual reinforcement.

distortions and discrepancies between the input texts and the generated videos, which hinders the overall quality and coherence of the generated content.

To address the above issues, we take one step further in this paper which highlights the complementary nature of both spatial and temporal features in videos. Specifically, we propose a novel Swapped spatiotemporal Cross-Attention (Swap-CA) for text-to-video generation. Instead of solely relying on separable 2D+1D self-attention (Bertasius et al., 2021) or 3D window self-attention (Liu et al., 2022) that replace computationally expensive 3D self-attention as shown in Fig. 1 (a)-(c), we aim to further enhance the interaction between spatial and temporal features. As shown in Fig. 1 (d), our swap attention mechanism facilitates bidirectional guidance between spatial and temporal features by considering one feature as the query and the other as the key/value. To ensure the reciprocity of information flow, we also swap the role of the "query" in adjacent layers.

By deeply interplaying spatial and temporal features through the proposed swap attention, we present a holistic VideoFactory framework for text-to-video generation. In particular, we adopt the latent diffusion framework and design a spatiotemporal U-Net for 3D noise prediction. To unlock the full potential of the proposed model and fulfill high-quality video generation, we construct the largest video generation dataset, named HD-VG-130M. This dataset consists of 130 million text-video pairs from open-domains, encompassing high-definition, widescreen, and watermark-free characters. Additionally, our spatial super-resolution model can effectively upsample videos to a resolution of $1376 \times 768$, thus ensuring engaging visual experience. We conduct comprehensive experiments and show that our approach outperforms existing methods in terms of both quantitative and qualitative comparisons. In summary, our paper makes the following significant contributions:

- We reveal the significance of learning joint spatial and temporal features for video generation, and introduce **a novel Swap-CA mechanism** to reinforce both space and time interactions. It significantly improves the generation quality, while ensuring precisely semantic alignment between the input text and the generated videos.

- We curate a comprehensive dataset comprising **the largest** 130 million text-video pairs to-date, which is **the first** to support high-quality video generation with high-definition, widescreen, and watermark-free characters. We believe this dataset will greatly benefit fellow researchers and advance the field of video generation.[1]

## 2 RELATED WORKS

**Text-to-Image Generation.** Generating realistic images from corresponding descriptions combines the challenging components of language modeling and image generation. Traditional text-to-image generation methods (Mansimov et al., 2016; Reed et al., 2016; Xu et al., 2018; Zhang et al., 2017) are mainly based on GANs (Goodfellow et al., 2014) and are only able to model simple scenes such as birds (Wah et al., 2011). Later work (Ramesh et al., 2021; Ding et al., 2021) extends the scope of text-to-image generation to open domains with better modeling techniques and training data on much

---

[1]Anonymous preview: Google Drive. It will be fully released upon the acceptance of our submission.

larger scales. In recent years, diffusion models have shown great ability in visual generation (Dhariwal & Nichol, 2021). For text-to-image multi-modality generation, GLIDE (Nichol et al., 2021), DALL·E 2 (Ramesh et al., 2022a), and Imagen (Saharia et al., 2022) leverage diffusion models to achieve impressive results. Based on these successes, some work extends latent diffusion (Rombach et al., 2022), customization (Ruiz et al., 2023), image guidance (Yang et al., 2023), and precise control (Balaji et al., 2022). This paper further extends diffusion models for video generation.

**Text-to-Video Generation.** Additional controls are often added to make the generated videos more responsive to demand (Mathieu et al., 2016; Pan et al., 2017; Wang et al., 2018), and this paper focuses on the controlling mode of texts. Early text-to-video generation models (Li et al., 2018; Pan et al., 2017) mainly use convolutional GAN models with Recurrent Neural Networks (RNNs) to model temporal motions. Although complex architectures and auxiliary losses are introduced, GAN-based models cannot generate videos beyond simple scenes like moving digits and close-up actions. Recent works extend text-to-video to open domains with large-scale transformers (Yu et al., 2022a) or diffusion models (Ho et al., 2022a). Considering the difficulty of high-dimensional video modeling and the scarcity of text-video datasets, training text-to-video generation from scratch is unaffordable. As a result, most works acquire knowledge from pretrained text-to-image models. CogVideo (Hong et al., 2022) inherits from a pretrained text-to-image model CogView2 (Ding et al., 2022). Imagen Video (Ho et al., 2022a) and Phenaki (Villegas et al., 2022) adopt joint image-video training. Make-A-Video (Singer et al., 2022) learns motion on video data alone, eliminating the dependency on text-video data. To reduce the high cost of video generation, latent diffusion has been widely utilized for video generation (An et al., 2023; Blattmann et al., 2023; Esser et al., 2023; He et al., 2022b;a; Khachatryan et al., 2023a; Ma et al., 2023; Wu et al., 2022; 2023; Yu et al., 2023; Zhou et al., 2022). MagicVideo (Zhou et al., 2022) inserts a simple adaptor after the 2D convolution layer. Latent-Shift (An et al., 2023) adopts a parameter-free temporal shift module to exchange information across different frames. PDVM (Yu et al., 2023) projects the 3D video latent into three 2D image-like latent spaces. Although the research on text-to-video generation is very active, existing research ignores the inter and inner correlation between spatial and temporal modules. In this paper, we revisit the design of text-driven video generation.

## 3 High-Definition Video Generation Dataset

Datasets of diverse text-video pairs are the prerequisite for training open-domain text-to-video generation models. However, existing text-video datasets are always limited in either scale or quality, thus hindering the upper bound of high-quality video generation. Referring to Tab. 1, MSR-VTT (Xu et al., 2016) and UCF101 (Soomro et al., 2012) only have 10K and 13K video clips respectively. Although large in scale, HowTo100M (Miech et al., 2019) is specified for instructional videos, which has limited diversity for open-domain generation tasks. Despite being appropriate in both scale and domain, the formats of textual annotations in HD-VILA-100M (Xue et al., 2022) are subtitle transcripts, which lack visual contents related descriptions for high-quality video generation. Additionally, the videos in HD-VILA-100M have complex scene transitions, which are disadvantageous for models to learn temporal correlations. WebVid-10M (Bain et al., 2021) has been used in some previous video generation works (Ho et al., 2022a; Singer et al., 2022), considering its relatively large-scale (10M) and descriptive captions. Nevertheless, videos in WebVid-10M are of low resolution and have poor visual qualities with watermarks in the center.

To tackle the problems above and achieve high-quality video generation, we propose a large-scale text-video dataset, namely **HD-VG-130M**, including 130M text-video pairs from open-domain in high-definition (720p), widescreen and watermark-free formats. We first collect high-definition videos from YouTube. The **challenge** lies in converting raw HD videos into video-caption pairs, which is far from straightforward. As the original videos have complex scene transitions which are adverse for models to learn temporal correlations, we detect and split scenes in these original videos using PySceneDetect (Breakthrough), resulting in 130M single scene video clips. Finally, we caption video clips with BLIP-2 (Li et al., 2023), in view of its large vision-language pre-training knowledge. To be specific, we extract the central frame in each clip as the keyframe, and get the annotation for each clip by captioning the keyframe with BLIP-2 (Li et al., 2023). Note that the video clips in HD-VG-130M are in single scenes, which ensures that the keyframe captions are representative enough to describe the content of the whole clips in most circumstances. The statistics of HD-VG-130M are shown in Fig. 2. The videos in HD-VG-130M cover 15 categories. The wide range of domains is beneficial for

Table 1: Comparison of different video datasets. Existing text-video datasets are always limited in either scale or quality, while our HD-VG-130M includes 130M text-video pairs from open-domain in high-definition, widescreen format, devoid of scene transitions, and free from watermarks. **Captions** are premium-quality text labels for videos. In contrast, class labels tend to be overly simplistic, and subtitles do not synchronize with the visual contents of the video.

| Dataset | Video clips | Resolution | Domain | Text | Transition-free | Watermark-free |
| --- | --- | --- | --- | --- | --- | --- |
| MSR-VTT (2016) | 10K | 240p | open | **caption** | ✓ | ✓ |
| UCF101 (2012) | 13K | 240p | human action | class label | ✓ | ✓ |
| HowTo100M (2019) | 136M | 240p | instructional | subtitle | ✗ | ✓ |
| HD-VILA-100M (2022) | 103M | 720p | open | subtitle | ✗ | ✓ |
| WebVid-10M (2021) | 10M | 360p | open | **caption** | ✓ | ✗ |
| HD-VG-130M (Ours) | 130M | 720p | open | **caption** | ✓ | ✓ |

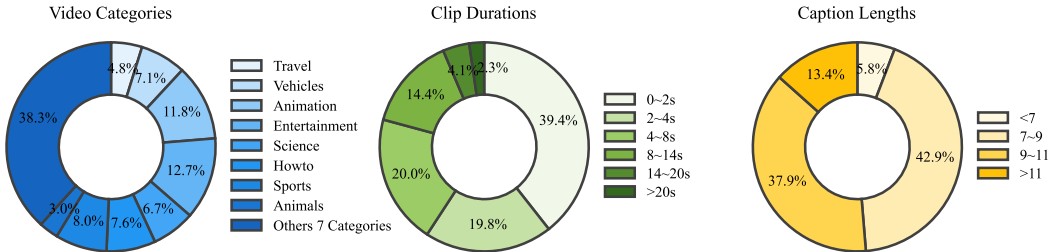

Figure 2: Statistics of video categories, clip durations, and caption word lengths in HD-VG-130M. HD-VG-130M covers a wide range of video categories.

training the models to generate diverse content. After scene detection, the video clips are mostly in single scenes with duration less than 20 seconds. The textual annotations are visual contents related to descriptive captions, which are mostly around 10 words. Text-video examples of our HD-VG-130M can be found in Figs. 9-10 in the *appendix*.

## 4 HIGH-QUALITY TEXT-TO-VIDEO GENERATION

**Spatiotemporal Inter-Connection.** To reduce computational costs and leverage pretrained image generation models, space-time separable architectures have gained popularity in text-to-video generation (Ho et al., 2022b; Hong et al., 2022). These architectures handle spatial operations independently on each frame, while temporal operations consider multiple frames for each spatial position. In the following, we refer to the features predicted by 2D/spatial modules in space-time separable networks as "spatial features", and "temporal features" vice versa.

The quality of spatiotemporal features is important for video generation, as it can affect temporal consistency and text-content alignment performance (Hong et al., 2022; Ho et al., 2022a). The interaction between spatial and temporal features is also essentially, as it determines how the spatial and temporal features are combined. This interaction has been highlighted in previous video-related studies (Bertasius et al., 2021; Zeng et al., 2020) and verified in cross-modality learning (Gu et al., 2023; Ruan et al., 2023). However, as discussed in Sec. 1, prior works have neglected the crucial interaction between spatial and temporal features. To tackle this limitation, we promote the mutual reinforcement of these features through a series of cross-attention operations.

Denote a basic operation $\text{CrossAttention}(x, y) = \text{softmax}(\frac{QK^T}{\sqrt{d}}) \cdot V$, with

$$Q = W_Q^{(i)} \cdot x, \quad K = W_K^{(i)} \cdot y, \quad V = W_V^{(i)} \cdot y, \tag{1}$$

where $W_Q^{(i)}$, $W_K^{(i)}$, and $W_V^{(i)}$ are learnable projection matrices in the $i$-th layer. The direction of cross-attention, specifically whether $Q$ originates from spatial or temporal features, plays a decisive role in determining the impact of cross-attention. In general, spatial features tend to encompass a greater amount of contextual information, which can improve the alignment of temporal features with the input text. On the other hand, temporal features have a complete receptive field of the time series, which may enable spatial features to generate visual content more effectively. To leverage

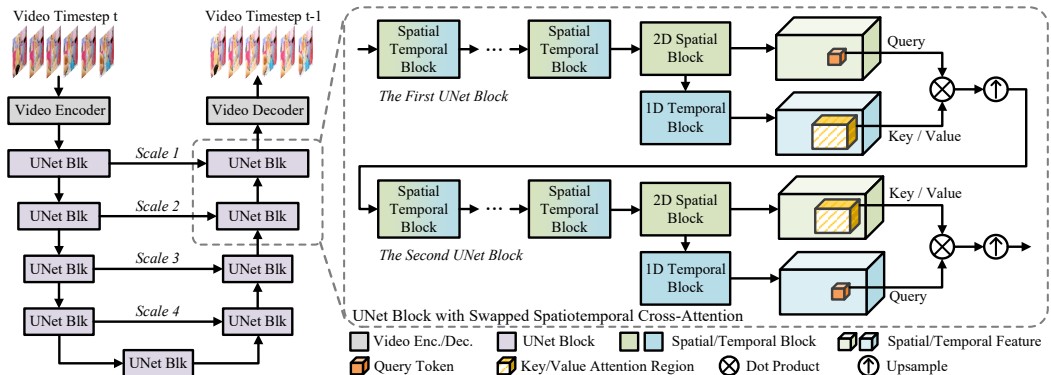

Figure 3: An illustration of our video diffusion model incorporating Swapped spatiotemporal Cross-Attention (Swap-CA). At the end of each U-Net block, we employ a swapped cross-attention scheme on 3D windows to facilitate a comprehensive integration of spatial and temporal features. In the case of two consecutive blocks, the first block employs temporal features to guide spatial features, while in the second block, their roles are reversed. This reciprocal arrangement ensures a balanced and mutually beneficial interaction between the spatial and temporal modalities throughout the model.

both aspects effectively, we propose a strategy of swapping the roles of $Q$ and $K, V$ in adjacent two blocks. This approach ensures that both temporal and spatial features receive sufficient information from the other modality, enabling a comprehensive and mutually beneficial interaction.

Global attention greatly increases the computational costs in terms of memory and running time. To improve efficiency, we conduct 3D window attention. Given a video feature in the shape of $F \times H \times W$ and a 3D window size of $F_w \times H_w \times W_w$, we organize the windows to process the feature in a non-overlapping manner, leading to $\lceil \frac{F}{F_w} \rceil \times \lceil \frac{H}{H_w} \rceil \times \lceil \frac{W}{W_w} \rceil$ distinct 3D windows. Within each window, we perform spatiotemporal cross-attention. By adopting the 3D window scheme, we effectively reduce computational costs without compromising performance.

Following prior text-to-image arts (Blattmann et al., 2023; Rombach et al., 2022), we incorporate $2\times$ down/upsampling along the spatial dimension to establish a hierarchical structure. Furthermore, research (Habibian et al., 2019; Pessoa et al., 2020) has pointed out that the temporal dimension is sensitive to compression. In light of these considerations, we do compress the temporal dimension and conduct shift windows (Liu et al., 2022), which advocates an inductive bias of locality. On the spatial dimension, we do not shift since the down/upsampling already introduces connections between neighboring non-overlapping 3D windows.

To this end, we propose a Swapped spatiotemporal Cross-Attention (Swap-CA) in 3D windows. Let $t^l$ and $s^l$ represent the predictions of 2D and 1D modules. We utilize Multi-head Cross Attention (MCA) to compute their interactions by Swap-CA as

$$
\begin{aligned}
\tilde{s}^l &= \text{Proj}_{in}^l \odot \text{GN}(s^l), \quad \tilde{t}^l = \text{Proj}_{in}^l \odot \text{GN}(t^l); \\
h^l &= \text{3DW-MCA}(\text{LN}(\tilde{s}^l), \ \text{LN}(\tilde{t}^l)) + \tilde{s}^l; \\
\bar{h}^l &= \text{FFN} \odot \text{LN}(h^l) + h^l; \\
z^l &= t^l + s^l + \text{Swap-CA}(s^l, t^l) = t^l + s^l + \text{Proj}_{out}^l(\bar{h}^l),
\end{aligned}
\tag{2}
$$

where GN, Proj, LN, 3D Window-based Multi-head Cross-Attention (3DW-MCA) are learnable modules. By initializing the output projection $\text{Proj}_{out}^{l-1}$ by zero, we have $z^l = t^{l-1} + s^{l-1}$, $i.e.$, Swap-CA is skipped so that it is reduced to a basic addition operation. This allows us to initially train the diffusion model using addition operations, significantly speeding up the training process. Subsequently, we can switch to Swap-CA to enhance the model's performance.

Then for the next spatial-temporal separable block, we apply shifted 3D window multi-head cross-attention (3DSW-MCA) and interchange the roles of $s$ and $t$, as

$$
h^{l+1} = \text{3DSW-MCA}(\text{LN}(\tilde{t}^{l+1}), \text{LN}(\tilde{s}^{l+1})) + \tilde{t}^{l+1}.
\tag{3}
$$

In all 3DSW-MCA, we shift the window along the temporal dimension by $\lceil \frac{F_w}{2} \rceil$ elements.

Table 2: Ablation study on spatiotemporal interaction strategies. We report the FVD (Unterthiner et al., 2018) and CLIPSIM (Radford et al., 2021) on 1K samples from the WebVid-10M (Bain et al., 2021) validation set. Computational cost is evaluated on inputs of shape $4 \times 16 \times 32 \times 32$. Details can be found in the *appendix*. $T$ and $S$ represent spatial and temporal features, respectively.

| Attention Type | $Q$ | $K, V$ | Param. (G) | Mem. (GB) | Time (ms) | FVD $\downarrow$ | CLIPSIM $\uparrow$ |
|---|---|---|---|---|---|---|---|
| - | - | - | 1.480 | 9.37 | 135.35 | 566.16 | 0.3070 |
| | $T$ | $S$ | 1.601 | 22.96 | 202.12 | 555.35 | 0.3091 |
| Global | $S$ | $T$ | 1.601 | 22.96 | 205.00 | 496.25 | 0.3073 |
| | Swapped | | 1.601 | 22.96 | 201.51 | 485.86 | 0.3092 |
| | $T$ | $S$ | 1.601 | 9.83 | 150.49 | 563.12 | 0.3086 |
| 3D Window | $S$ | $T$ | 1.601 | 9.83 | 149.93 | 490.60 | 0.3076 |
| | Swapped | | 1.601 | 9.83 | 148.24 | **475.09** | **0.3107** |

**Overall Architecture.** We adopt LDM (Rombach et al., 2022) as the text-to-image backbone. We employ an auto-encoder to compress the video into a down-sampled 3D latent space. Within this latent space, we perform diffusion optimization using an hourglass spatial-temporal separable U-Net model. Text features are extracted with a pretrained CLIP (Radford et al., 2021) model and inserted into the U-Net model through cross-attention on the spatial dimension.

Our framework is illustrated in Fig. 3. To balance performance and efficiency, we use Swap-CA only at the end of each U-Net encoder and decoder block. In other positions, we employ a straightforward fusion technique using a $1 \times 1 \times 1$ convolution to merge spatial and temporal features. To enhance the connectivity among temporal modules, we introduce skip connections that connect temporal modules separated by spatial down/upsampling modules. This strategy promotes stronger integration and information flow within the temporal dimension of the network architecture.

**Super-Resolution Towards Higher Quality.** To obtain visually satisfying results, we further perform Super-Resolution (SR) on the generated video. One key to improving SR performance is designing a degradation model that closely resembles the actual degradation process (Wang et al., 2021). In our scenario, the generated video quality suffers from both the diffusion and auto-encoder processes. Therefore, we adopt the hybrid degradation model in Real-ESRGAN (Wang et al., 2021) to simulate possible quality degradation caused by the generated process. During training, an original video frame is downsampled and degraded using our model, and the SR network attempts to perform SR on the resulting low-resolution image. We adopt RCAN (Zhang et al., 2018) with 8 residual blocks as our SR network. It is trained with a vanilla GAN (Goodfellow et al., 2014) to improve visual satisfaction. With a suitable degradation design, our SR network can further reduce possible artifacts and distortion in the frames, increase their resolution, and improve their visual quality.

## 5 EXPERIMENTS

### 5.1 IMPLEMENTATION DETAILS

Our model predicts images at a resolution of $344 \times 192$ (with a latent space resolution of $43 \times 24$). Then a $4\times$upscaling is produced in our SR model, resulting in a final output resolution of $1376 \times 768$. Our model is trained with 32 NVIDIA V100 GPUs. We utilize our HD-VG-130M as training data to promote the generation visual qualities. Furthermore, considering that the textual captions in HD-VG-130M are annotated by BLIP-2 (Li et al., 2023), which may have some discrepancies with human expressions, we adopt a joint training strategy with WebVid-10M (Bain et al., 2021) to ensure the model could generalize well to diverse humanity textual inputs. This approach allows us to benefit from the large-scale text-video pairs and the superior visual qualities of HD-VG-130M while maintaining the generalization ability to diverse textual inputs in real scenarios, enhancing the overall training process. More details can be found in the *appendix*.

### 5.2 ABLATION STUDIES

**Spatiotemporal Inter-Connection.** We first evaluate the design of our swapped cross-attention mechanism. As shown in Tab. 2, using temporal features as $Q$ generally leads to better CLIP similarity

Figure 4: Generation results w/o and w/ HD-VG-130M for training.

Figure 5: Training on de-watermarked WebVid-10M.

Figure 6: Different video captioning results.

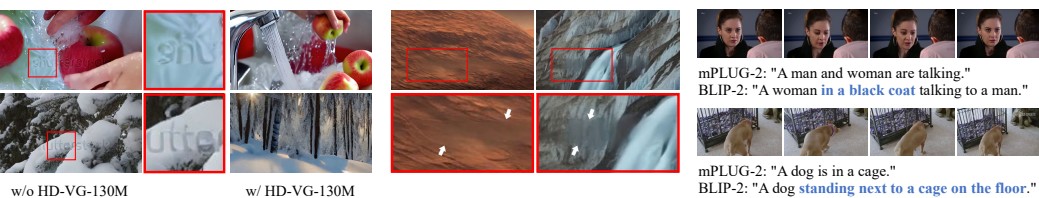

w/o HD-VG-130M     w/ HD-VG-130M

mPLUG-2: "A man and woman are talking."
BLIP-2: "A woman **in a black coat** talking to a man."

mPLUG-2: "A dog is in a cage."
BLIP-2: "A dog **standing next to a cage on the floor**."

(CLIPSIM) (Radford et al., 2021), revealing a better text-video alignment. The reason might be that language cross-attention only exists in spatial modules. Thus, using spatial features to guide temporal ones implicitly enhance semantic guidance. Reversely, using spatial as $Q$ leads to significantly better FVD, revealing better video quality. The reason might be that the spatial features can better perceive the overall video by using temporal fea-

Table 3: Ablation study on attention strategies.

| Methods | FVD $\downarrow$ | CLIPSIM $\uparrow$ |
|---|---|---|
| Baseline | 566.16 | 0.3070 |
| Tune-A-Video (2023) | 717.34 | 0.3084 |
| CogVideo (2022) | 534.48 | 0.3010 |
| 3D Spatial-Temporal WSA | 500.49 | 0.3072 |
| Swap-CA (Ours) | **475.09** | **0.3107** |

tures as guidance. This experiment demonstrates the benefits of introducing cross-attention, as well as the different acts of spatial and temporal features. Combining these two aspects, we propose to swap the roles of $x$ and $y$ every two blocks. In this way, both the temporal and spatial features can get sufficient information from the other modality, leading to improved FVD and CLIPSIM scores. 3D window attention not only significantly lowers computational costs but also leads to a slight performance improvement. Previous studies (Li et al., 2021; Wang et al., 2022) have observed similar performance improvements by integrating a module to enhance local information within transformer-like structures.

We conduct comparisons with other attention strategies in Tab. 3. We re-implement these designs within our framework. Specifically, 3D spatial-temporal WSA is realized by first adding spatial and temporal features together and then applying 3D window self-attention. All other settings remain consistent with Tab. 2. The custom attention mechanism utilized in the one-shot model, Tune-A-Video, appears to be less effective in the open-domain setting. While CogVideo and 3D spatial-temporal WSA surpass the baseline, they bring less performance improvement compared with our Swap-CA, showing the effectiveness of our approach.

For additional subjective results and evaluation regarding different window sizes, please consult Sec. E.1 and Sec. E.2 in the *appendix* due to space limitations.

**High-Definition Video Generation Dataset.** The advantages of HD-VG-130M extend beyond watermark removal. As shown in Fig. 4, training with HD-VG-130M not only eliminates watermarks but also elevates the scenic beauty and enriches the level of detail, leading to a comprehensive improvement in the visual quality of the generated videos. This improvement is further reflected in a 10% decrease in FVD: 475.09 for "w/o HD-VG-130M" and 429.75 for "w/ HD-VG-130M".

We also tried using E2FGVI (Li et al., 2022) to remove watermarks from WebVid-10M. As shown in Fig. 5, the generated videos have blurry textures. Removing watermarks from WebVid-10M to produce high-quality video data is non-trivial, which reveals the significance of our HD-VG-130M.

We further evaluated different captioning models. We experimented with a state-of-the-art video captioning model, mPLUG-2 (Xu et al., 2023), but observed that it provides less detailed descriptions (e.g., BLIP-2 predicts "black coat" while mPLUG-2 does not in the first row of Fig. 6) or misinterprets the scene (e.g., mistakes the dog to be inside the cage in the second row of Fig. 6). As a result, using videos captioned with mPLUG-2, the CLIPSIM is decreased to 0.3046.

Finally, we assessed the impact of training with HD-VILA-100M (Xue et al., 2022) instead of HD-VG-130M. As HD-VILA-100M only provides subtitles and lacks scene detection (with potential multiple transitions), significant performance degradation is observed in FVD (429.75 → 692.99) and CLIPSIM (0.3082 → 0.2671), despite joint training with WebVid. This experiment highlights the crucial role of our scene detection and video captioning procedures.

Table 4: Text-to-video generation on UCF101. Our VideoFactory surpasses the SoTA method (Video LDM) by 25%.

| Method | Zero-shot | FVD↓ |
|---|---|---|
| VideoGPT (2021) | No | 2880.6 |
| MoCoGAN (2018) | No | 2886.8 |
| +StyleGAN2 (2020) | No | 1821.4 |
| MoCoGAN-HD (2021) | No | 1729.6 |
| DIGAN (2022b) | No | 1630.2 |
| StyleGAN-V (2022) | No | 1431.0 |
| PVDM (2023) | No | 343.6 |
| CogVideo (2022) | Yes | 701.6 |
| MagicVideo (2022) | Yes | 699.0 |
| LVDM (2022b) | Yes | 641.8 |
| ModelScope (2023) | Yes | 639.9 |
| Video LDM (2023) | Yes | 550.6 |
| Ours | Yes | **410.0** |

Table 5: Text-to-video generation on MSR-VTT.

| Method | Zero-shot | CLIPSIM↑ |
|---|---|---|
| GODIVA (2021) | No | 0.2402 |
| NUWA (2022) | No | 0.2439 |
| LVDM (2022b) | Yes | 0.2381 |
| CogVideo (2022) | Yes | 0.2631 |
| ModelScope (2023) | Yes | 0.2795 |
| Video LDM (2023) | Yes | 0.2929 |
| Ours | Yes | **0.3005** |

Table 6: Text-to-video generation on WebVid. We beat the SoTA method by 29% in FVD.

| Method | FVD↓ | CLIPSIM↑ |
|---|---|---|
| LVDM (2022b) | 455.53 | 0.2751 |
| ModelScope (2023) | 414.11 | 0.3000 |
| Ours | **292.35** | **0.3070** |

Table 7: User Preference. The number indicates the percentage of humans that prefer our method over the compared method. We also show the ratio of the network parameter v.s. Ours.

| Sample | Method | Param Ratio | Video Quality | Text-Video | Overall |
|---|---|---|---|---|---|
| Pretrained Model | ModelScope (Luo et al., 2023) | 0.90× | 0.8875 | 0.8575 | 0.9300 |
| | LVDM (He et al., 2022b) | 0.57× | 0.9155 | 0.8555 | 0.9370 |
| Open Website | Make-A-Video (Singer et al., 2022) | 4.76× | 0.5417 | 0.4958 | 0.5417 |
| | Imagen Video (Ho et al., 2022a) | 7.97× | 0.4291 | 0.2582 | 0.3818 |

## 5.3 QUANTITATIVE AND QUALITATIVE COMPARISON

To thoroughly evaluate the generation performance of our VideoFactory, we benchmark it on three distinct datasets: WebVid-10M (Bain et al., 2021) (Val) same as the domain of part of our training data, along with UCF101 (Soomro et al., 2012) and MSR-VTT (Xu et al., 2016) in zero-shot setting.

**Evaluation on UCF101.** As mentioned in Sec. 3, the textual annotations in UCF101 are class labels. We first follow (Ho et al., 2022b; Singer et al., 2022) and rewrite the labels of 101 classes to descriptive captions, and then generate 100 samples for each class. As shown in Tab. 4, the FVD of our methods reaches 410.0, which achieves the best compared with other methods both in zero-shot setting and beats most of the methods which have tuned on UCF101. The results verify that our proposed VideoFactory could generate more coherent and realistic videos.

**Evaluation on MSR-VTT.** As shown in Tab. 5, we also evaluate the CLIPSIM on the widely used video generation benchmark MSR-VTT. We randomly choose one prompt per example from MSR-VTT to generate 2990 videos in total. Although in a zero-shot setting, our method achieves the best compared to other methods with an average CLIPSIM score of 0.3005, which suggests the semantic alignment between the generated videos and the input text.

**Evaluation on WebVid-10M (Val).** Referring to Tab. 6, we randomly extract 5K text-video pairs from WebVid-10M which are exclusive from the training data to form a validation set and conduct evaluations on it. Our approach achieves an FVD of 292.35 and a CLIPSIM of 0.3070, outperforming existing methods and showcasing the superiority of our approach.

**Human Evaluation.** To assess our VideoFactory from the aspect of humans, we conducted a user study comparing it with four state-of-the-art models. Specifically, we selected two models, ModelScope and LVDM, for which codes and pretrained models were available, and two methods, Make-A-Video and Imagen Video, which only show some samples on their websites. In each case, participants were presented with two samples generated by our method and the competitor. They were then asked to evaluate the video quality, text-video correlation, and express an overall preference. The results are summarized in Tab. 7. Additionally, we provide the parameter ratios for fair comparisons.

**Subjective Results.** In Fig. 7, we show comparison results against Make-A-Video, Imagen Video, and Video LDM. The prompts and generated results are collected from their official project website. We

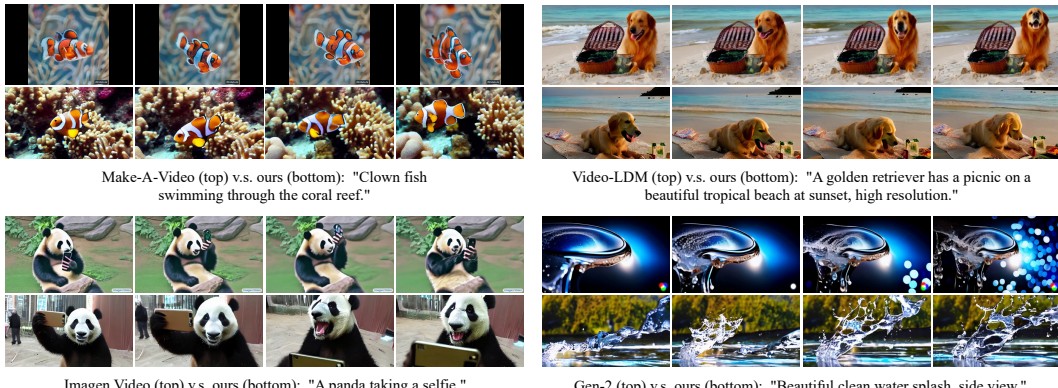

Make-A-Video (top) v.s. ours (bottom): "Clown fish swimming through the coral reef."

Video-LDM (top) v.s. ours (bottom): "A golden retriever has a picnic on a beautiful tropical beach at sunset, high resolution."

Imagen Video (top) v.s. ours (bottom): "A panda taking a selfie."

Gen-2 (top) v.s. ours (bottom): "Beautiful clean water splash, side view."

Figure 7: Text-to-video generation results compared with Make-A-Video, Imagen Video, Video-LDM, and Gen-2 (Cases of the first three methods are collected from their public project websites).

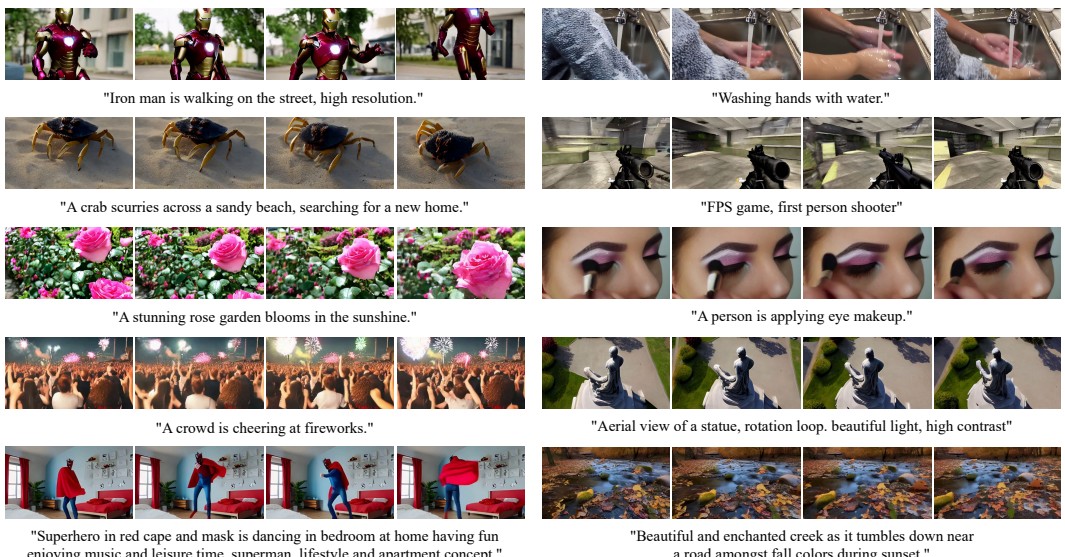

"Iron man is walking on the street, high resolution."

"Washing hands with water."

"A crab scurries across a sandy beach, searching for a new home."

"FPS game, first person shooter"

"A stunning rose garden blooms in the sunshine."

"A person is applying eye makeup."

"A crowd is cheering at fireworks."

"Aerial view of a statue, rotation loop. beautiful light, high contrast"

"Superhero in red cape and mask is dancing in bedroom at home having fun enjoying music and leisure time. superman, lifestyle and apartment concept."

"Beautiful and enchanted creek as it tumbles down near a road amongst fall colors during sunset."

Figure 8: Samples generated by our VideoFactory exhibit high quality, featuring clear motion, intricate details, and precise semantic alignment.

also evaluate Gen-2 (Runway), a popular platform in the AIGC field. Make-A-Video only generates 1:1 videos, which limits the user experience. When compared with Imagen Video and Video LDM, our model generates the panda and golden retriever with more vivid details. Despite setting the motion intensity parameter to the maximum, Gen-2 cannot simulate the splashing motion of water. Furthermore, we showcase additional samples of our model in Fig. 8 and Sec. E.4 in the *appendix*. **Video demos can be found in our supplementary.**

Due to space constraints, please refer to Sec. E.3 for failure case study.

## 6 CONCLUSION

In this paper, we introduce VideoFactory, a high-quality open-domain video generation framework that produces watermark-free, high-definition (1376×768), widescreen (16:9) videos. We enhance spatial and temporal modeling using a novel swapped cross-attention mechanism, allowing spatial and temporal information to complement each other effectively. Additionally, we provide the HD-VG-130M dataset, featuring 130 million open-domain text-video pairs in widescreen, watermark-free, high-definition format, maximizing the potential of our model. Experimental results demonstrate that VideoFactory generates videos with superior spatial quality, temporal consistency, and alignment with text, establishing it as the new benchmark for text-to-video generation.

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

## APPENDIX A    VIDEOS

For playable videos, we refer the reader to the video file available at ICLR-VideoFactory.mp4. This video file visually demonstrates the outputs and results discussed in this document.

## APPENDIX B    BROADER IMPACT AND LIMITATIONS

The proposed text-to-video generation model has broader implications and potential impact in various domains. It can benefit content creators, filmmakers, and video production companies, enabling them to automate the process of generating high-quality videos from textual descriptions. This technology also has the potential to enhance storytelling in multimedia platforms and improve accessibility for individuals with visual impairments.

However, it is essential to acknowledge certain limitations of the model. Firstly, the model's performance heavily relies on the quality and diversity of the training data. Although we have taken steps to process LAION, WebVid, and our proposed HD-VG by implementing NSFW (Not Safe for Work) filtering, there is still a possibility that biases and limitations may persist in the data, which can subsequently affect the generated videos. It is important to remain vigilant and continually improve data curation techniques to mitigate such biases and ensure the fairness and inclusivity of the generated content.

Moreover, while the model can generate visually compelling videos, it may struggle with complex or abstract textual descriptions. Certain nuanced concepts or creative elements may be challenging for the model to accurately represent in the generated videos. Continued research and development are required to improve the model's understanding of context and subtleties in text-to-video generation.

Furthermore, ethical considerations surrounding the use of generated videos should be taken into account. Misuse or malicious manipulation of this technology could potentially lead to the creation of deep fake videos or misinformation campaigns. Robust security measures and awareness of the potential risks are vital to mitigate these challenges.

Overall, the proposed text-to-video generation model offers exciting possibilities but requires ongoing scrutiny, responsible deployment, and further advancements to address limitations and ensure positive societal impacts.

## APPENDIX C    DATASETS

### C.1    HD-VG

We created a large dataset called HD-VG-130M. This dataset contains 130 million pairs of text and corresponding videos. The videos in this dataset are of high-definition quality, widescreen format and do not contain any watermarks or additional characters. It contains open-domain videos in 15 categories. Detailed category statistics are shown in Tab. 8.

Table 8: Detailed category statistics of HD-VG-130M.

| Category | Music | Nonprofits | Animals | Travel | Gaming | Comedy | Science |
|---|---|---|---|---|---|---|---|
| Percentage | 1.06% | 1.83% | 2.97% | 4.83% | 5.24% | 5.97% | 6.73% |
| Education | Vehicles | Howto | Blogs | Sports | News | Animation | Entertainment |
| 6.76% | 7.10% | 7.61% | 7.89% | 7.96% | 9.57% | 11.81% | 12.67% |

In constructing our dataset, we exploit the HD-VILA-100M dataset (Xue et al., 2022) to streamline the video acquisition process. We employ the video labels provided by HD-VILA-100M to selectively gather original high-definition videos from YouTube. Subsequently, we apply scene detection and captioning techniques to process the raw video content.

Examples of our dataset are shown in Figs. 9-10. By utilizing this dataset, we aim to maximize the capabilities of our model in generating superior videos.

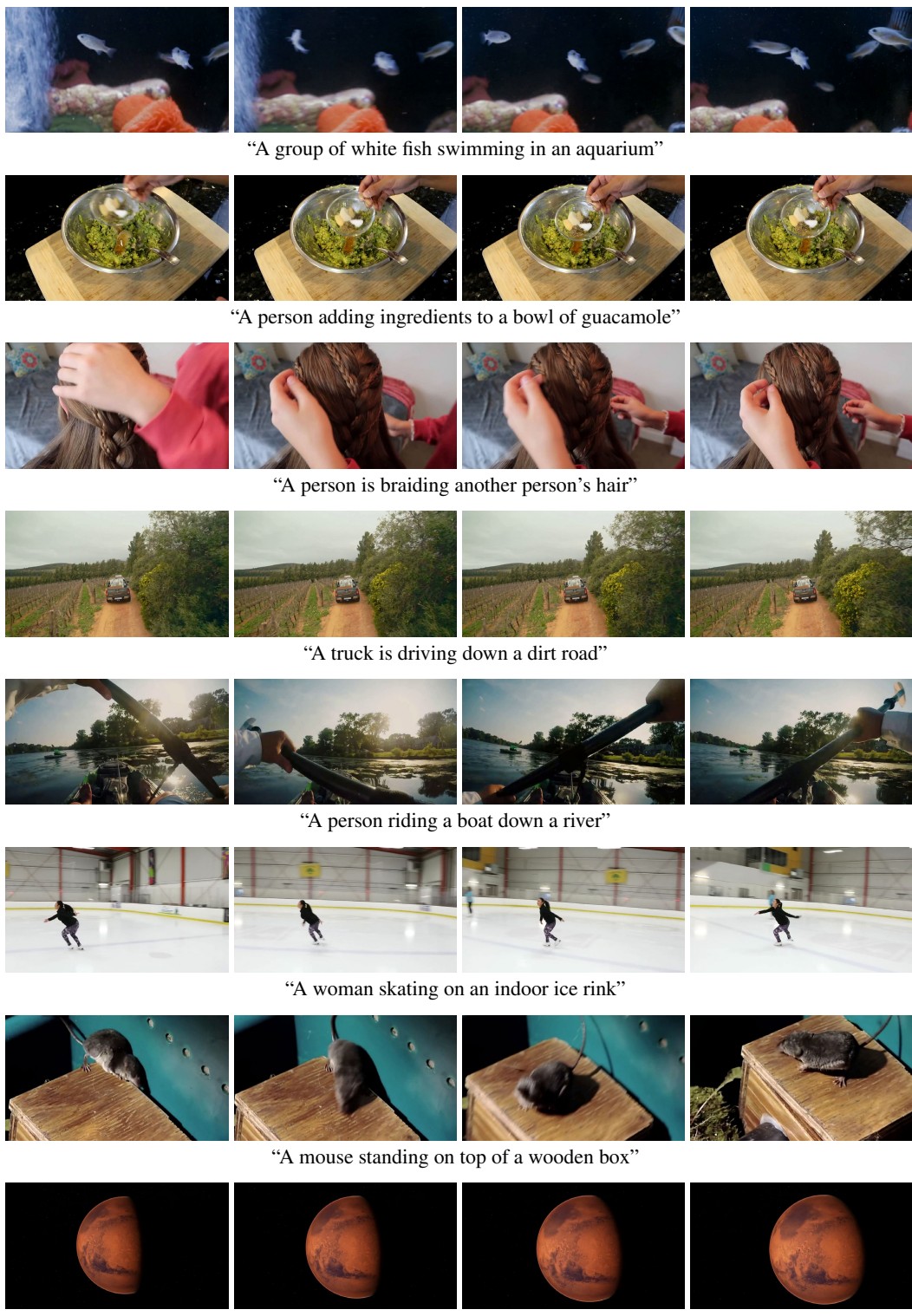

Figure 9: Video-caption paired samples in our HD-VG-130M dataset.

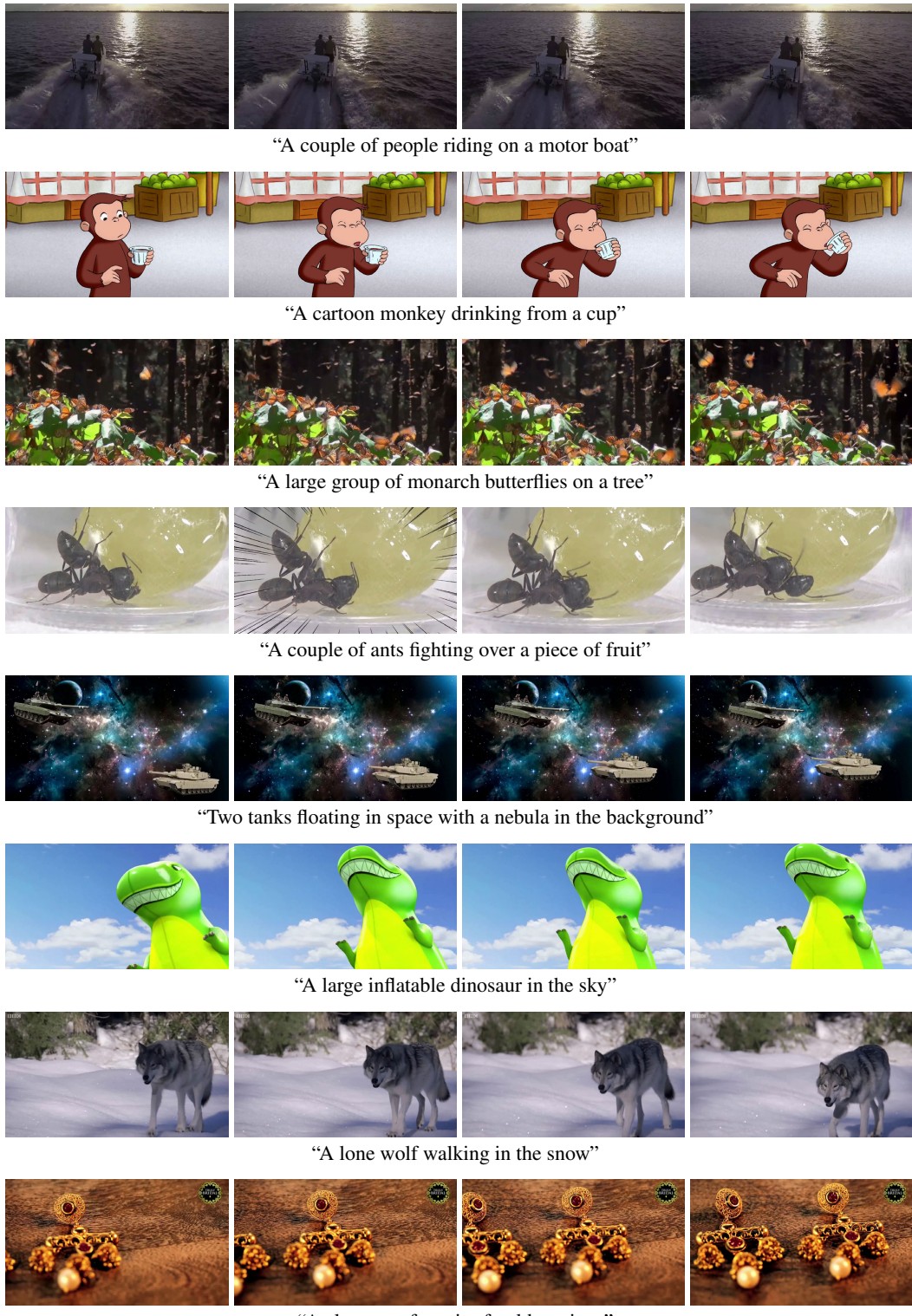

Figure 10: Video-caption paired samples in our HD-VG-130M dataset.

## C.2 WEBVID

WebVid is a comprehensive dataset consisting of short videos paired with textual descriptions. The dataset comprises a total of 10.7 million video-caption pairs, which correspond to approximately 52k hours. We utilize the 10 million samples from the WebVid-10M training set during our training process. In the quantitative comparison experiment, we employ the 5k samples from the validation set. In the ablation study, we randomly select 1,000 samples from the validation set for convenience.

## C.3 LAION

In addition to the aforementioned datasets, we utilize the LAION-Aesthetics subset for our joint image-video training. When processing image data, we skip the temporal layers by turning off the skip connections. Furthermore, to ensure widescreen compatibility, we filter out images with an aspect ratio smaller than 4:3. This step helps to maintain consistency and optimize the training process specifically for widescreen video generation.

# APPENDIX D   IMPLEMENTATION DETAILS

## D.1 TRAINING

To effectively adapt the pretrained parameters from a model trained in 1:1 ($256{\times}256$) aspect ratio to generating videos in 16:9, our model predicts images at a resolution of $344{\times}192 \approx 256{\times}256$ (with a latent space resolution of $43{\times}24$), as we found that generation models are sensitive to the areas of the generated images. Then a $4{\times}$upscaling is produced in our SR model, resulting in a final output resolution of $1376{\times}768$.

We initialize our model by partially loading usable pre-trained parameters obtained from ModelScope (Luo et al., 2023). To initialize new parameters, we employ zero convolution (Zhang & Agrawala, 2023) so that the new layers do not affect the loaded pretraining layer parameters at the initial stage of training. First of all, we train the model on WebVid, HD-VG, and LAION datasets for 18k steps without utilizing the proposed Swap-CA. After this initial training, the model still generates watermarks with a certain probability. Next, we solely train the model on the HD-VG and LAION dataset for 10k steps. Finally, we introduce the Swap-CA technique and continue training for an additional 12.5k steps. Throughout the training process, we use a batch size of 128. The model is trained using 32 V100 GPUs (4 batches per GPU), and the entire training process takes approximately 80 V100 days.

The learning rate is set to 1e-5. We use the AdamW (Loshchilov & Hutter, 2019) optimizer with the following settings: $\beta_1 = 0.9$, $\beta_2 = 0.9$, $\lambda = 0.3$ (weight decay).

We utilize the sampler from Denoising Diffusion Implicit Models (DDIM) (Song et al., 2021) with 1000 steps, $\beta_0 = 0.00085$, and $\beta_T = 0.0120$. The scheduler for $\beta$ follows a linear scaling approach. For classifier-free guidance, the conditioning dropout rate $p_{\text{uncond}} = 0.2$.

## D.2 SAMPLING

Following the training settings, the resolution of the latent space is $43{\times}24$, which is decoded into $344{\times}192$ by the decoder in the auto-encoder, and finally upsampled into $1376{\times}768$ by the SR model.

We employ the sampler from DDIM (Song et al., 2021) with the following settings: 50 steps, $\beta_0 = 0.00085$, and $\beta_T = 0.0120$. The scheduler of $\beta$ is scaled linear. In classifier-free guidance (Ho & Salimans, 2022), we set the guidance scale to 9.0 following ModelScope (Luo et al., 2023).

## D.3 QUANTITATIVE EVALUATION

For the quality of video generation, we evaluate Fréchet Video Distance (FVD) (Unterthiner et al., 2018) on UCF101 (Soomro et al., 2012) and WebVid (Bain et al., 2021) datasets. For text-to-video alignment, we compute CLIP Similarly scores (CLIPSIM) (Radford et al., 2021) on MSR-VTT (Xu et al., 2016) and WebVid (Bain et al., 2021) datasets. We also conduct a user study to evaluate from the aspect of humans.

**FVD** quantifies the similarity between real and generated videos. For UCF101, we first generate 10k samples following (Singer et al., 2022). The prompts we use to generate samples are:

- A person is applying eye makeup.
- A person is applying lipstick.
- A person is doing archery.
- A baby is crawling.
- A player standing on balancebeam.
- A band marching on the street.
- A person pitch a baseball.
- Basketball players play basketball.
- Basketball player dunk a basketball.
- A person is doing bench press.
- A person is biking
- People playing billiards
- A person is blowing hairs with a blow dryer in hand.
- A person blowing candles.
- A person is doing body weight squats.
- A person is playing bowling.
- A person is boxing with a punching bag.
- A person is boxing with a speed bag.
- A person is doing breast stroke.
- A person is brushing teeth.
- A person is doing clean and jerk.
- A person Diving from a cliff.
- A person play cricket bowling.
- A person shot a cricket.
- A person is cutting in kitchen.
- A person is diving.
- A person is drumming.
- Two people are fencing
- A player is doing a field hockey penalty.
- A player doing floor gymnastics.
- A people catch a frisbee.
- A person is doing front crawl.
- A person is doing golf swing .
- A person got his hair cut by another person.
- A person is hammering.
- A person is throwing a hammer.
- A person is doing hand stand push ups.
- A person is doing hand stand walking.
- A person is getting head massage by another person.

- A person is high jumping.
- People riding horses in a horse-race.
- A person is riding a horse.
- A person playing with a hula hoop.
- People perform ice dancing.
- A person throw a javelin.
- A person play with juggling balls.
- A person is doing jumping Jack.
- A person is jumping rope.
- A person is kayaking.
- A person is knitting.
- A person is doing long jump.
- A person is doing lunges.
- A group of soldiers perform a military parade.
- A person is mixing ingredients in a bowl with a whisk.
- A person is mopping floor.
- A person play with nun chucks.
- A person take exercise on parallel bars.
- A person is tossing a pizza.
- A person is playing cello.
- A person is playing a daf.
- A person is playing a dhol.
- A person is playing a flute.
- A person is playing guitar.
- A person is playing piano.
- A person is playing sitar.
- A person is playing tabla.
- A person is playing violin.
- A person is pole-vaulting.
- A person is performing pommel horse.
- A person is performing pull ups.
- Two players confront in a punching match.
- A person is doing push ups.
- A person is Rafting.
- A person is rock climbing in door.
- A person is rope climbing.
- A person is rowing boat.
- people is performing Salsa Spin.
- A man is shaving his Beard.
- A player throw a shot-put.
- A person is playing skateboarding.

- A person is skiing.
- A person riding a skijet
- people perform sky-diving.
- A person is performing soccer juggling.
- A person is doing soccer penalty.
- A player is doing still rings.
- people are sumo wrestling on a match.
- A person is surfing.
- Children playing on the swing.
- A player shoot a table tennis.
- A person is doing tai chi.

- A tennis player swing a tennis racket.
- A player throw a discus.
- A player perform trampoline jumping.
- A person is typing on a computer.
- A player is doing gymnastics on an uneven bars.
- A player spiking a volley ball.
- A person is walking with a dog.
- A person is doing wall push ups.
- A person is writing on board.
- A person is playing YoYo.

For WebVid, we use the captions in its validation set. Next, we extract features from a pre-trained I3D action classification model. To establish reference statistics, we select video sequences from the dataset that consist of at least 16 frames. To eliminate randomness, we clip the central 16 frames of each sequence. Our implementation is based on the code provided by (Yan et al., 2021)[2].

**CLIPSIM** quantifies the alignment of videos to texts. The test set of MSR-VTT (Xu et al., 2016) contains 2990 videos with 20 descriptions per video. We randomly choose one prompt per example (random seed is set to 42). For WebVid, we generate samples based on the captions provided in its validation set. We use the ViT-B/32 model to compute the CLIP score. Our implementation is based on the code provided by (Radford et al., 2021)[3].

**User Study** includes four compared methods. Regarding ModelScope (Luo et al., 2023) and LVDM (He et al., 2022b), our model is compared to them using a sample of 100 randomly selected prompts from the validation set of WebVid (Bain et al., 2021). For Make-A-Video (Singer et al., 2022), our model is compared to 40 prompts, and the results of Make-A-Video are from its official website[4]. Similarly, for Imagen Video (Ho et al., 2022a), our model is compared to 55 prompts, and the results are compared with the outputs available on the official website of Imagen Video[5].

During the evaluation process, each participant will receive two samples of the same text: one generated by our method and the other by a competitor. They are then asked to compare the two samples based on video quality, text-video correlation, and overall preference.

To quantify the user preferences, we calculate the user preference ratio. The user preference ratio is calculated by comparing the number of participants who prefer the output from our method to the total number of participants. A ratio of 1 indicates that our method is preferred by all participants, 0.5 represents a comparable preference between our method and the competitor, and 0 indicates that the competitor is preferred by all participants. This ratio serves as a measure of the relative preference for our method compared to the competitor.

## APPENDIX E  ADDITIONAL RESULTS

### E.1  SUBJECTIVE ABLATION STUDY RESULTS

We show comparative examples in Fig. 11. These results demonstrate how our cross-attention design distinctly enhances scene quality and video dynamics. Furthermore, leveraging HD-VG-130M during training enhances scene aesthetics and object stability.

---

[2]https://github.com/wilson1yan/VideoGPT/

[3]https://huggingface.co/docs/transformers/model_doc/clip

[4]https://make-a-video.github.io

[5]https://imagen.research.google/video/

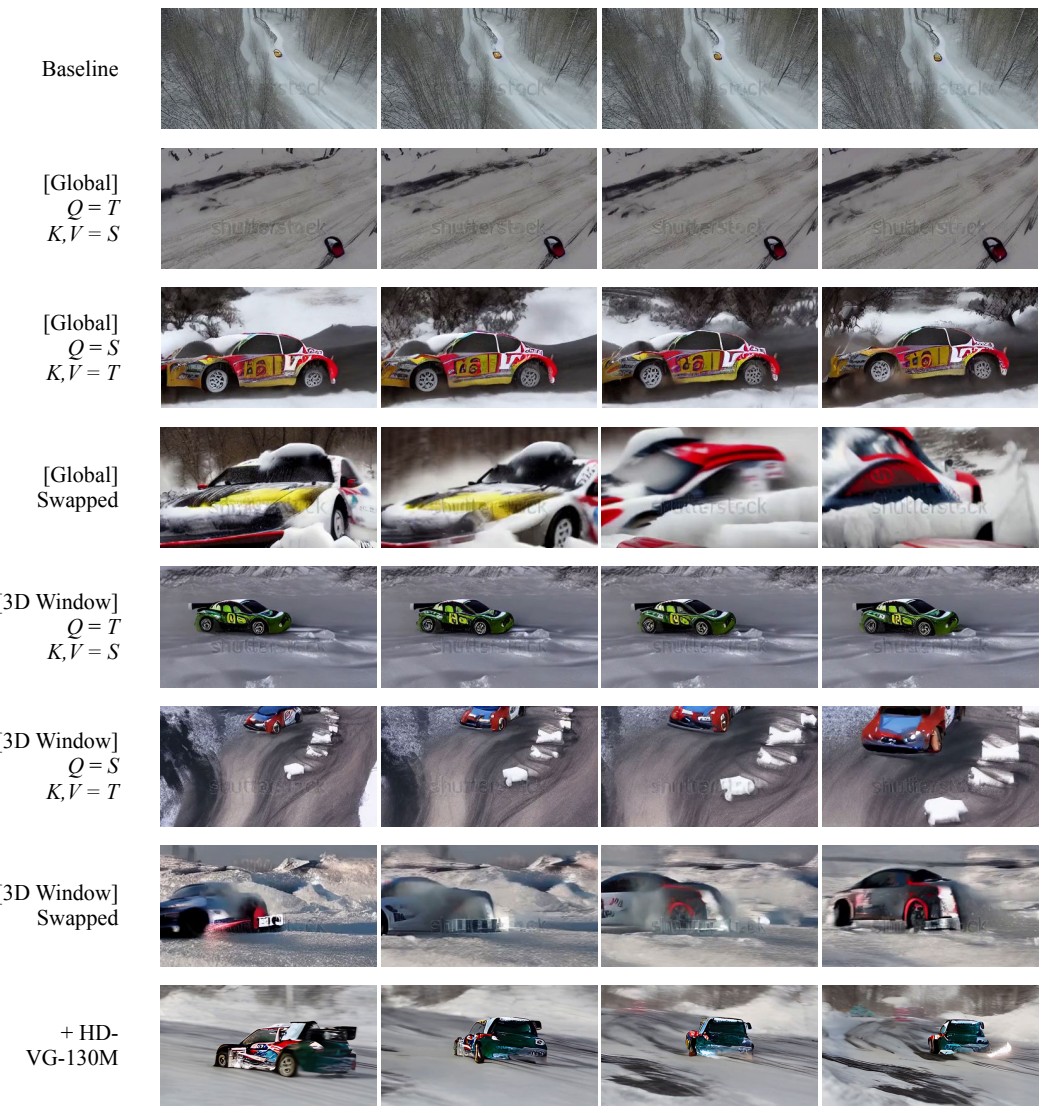

"Rally racing car ice racing, realistic"

Figure 11: Subjective ablation study results.

## E.2 ABLATION STUDY ON WINDOW SIZE

The final window size is set to $8 \times 3 \times 6$, *i.e.*, $F_w = 8$, $H_w = 3$, and $W_w = 6$. The rationale behind choosing $H_w = 3$ and $W_w = 6$ is to match the spatial resolution of the core feature in U-net, ensuring that the window attention in the core block can fully perceive the video contents. As for $F_w$, we set it to 8 to achieve a broader temporal attention view while reducing computation complexity.

Tab. 9 shows the ablation study we performed on window sizes, following the experimental setup in Tab. 2 of our main paper. Due to an NVIDIA software upgrade, we re-run the experiment, so the memory values are not the same in Tab. 2 and Tab. 9. Our final configuration, $8 \times 3 \times 6$, achieves the best FVD and CLIPSIM scores, while also delivering comparable efficiency.

Table 9: Ablation study on attention window size.

| Window Size ($F_w \times H_w \times W_w$) | Param. (G) | Mem. (GB) | Time (ms) | FVD $\downarrow$ | CLIPSIM $\uparrow$ |
|---|---|---|---|---|---|
| $8 \times 1 \times 3$ | 1.601 | 10.07 | 149.42 | 525.91 | 0.3056 |
| $4 \times 3 \times 6$ | 1.601 | 10.07 | 152.14 | 485.43 | 0.3064 |
| $8 \times 3 \times 6$ (Final Setting) | 1.601 | 10.07 | 153.16 | 475.09 | 0.3107 |
| $16 \times 3 \times 6$ | 1.601 | 10.07 | 153.23 | 487.08 | 0.3072 |
| Global Attention | 1.601 | 23.51 | 205.58 | 485.86 | 0.3092 |

## E.3 FAILURE CASE STUDY

The typical failure case of our text-to-video generation model is that our text encoder, CLIP (Radford et al., 2021), can sometimes misinterpret concepts, leading to unintended results. For instance, with the input prompt "A cat singing in a barbershop quartet," the term "barbershop quartet" signifies musical performance in a specific style. However, our text encoder might inadvertently emphasize "barbershop", introducing a corresponding background to the video. To address this, we can use GPT-3.5 for prompt refinement, after which our model can generate a vivid cat singing on the stage. Please refer to Fig. 12 for illustration.

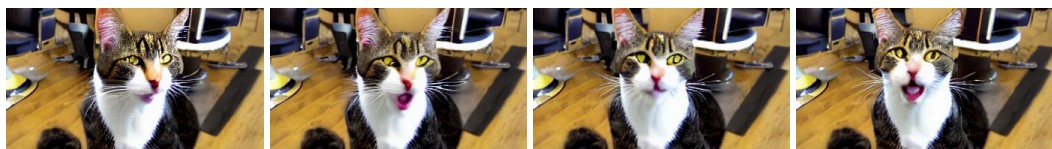

Original prompt: "A cat singing in a barbershop quartet. 4k HD, vivid"

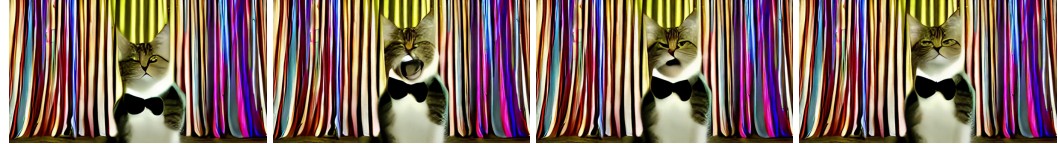

Revised prompt: "A cat singing in stage, wearing a bow tie. The background is vertical striped curtain. 4k HD, vivid."

Figure 12: Failure case study of our text-to-video generation model.

Another notable failure case arises from the presence of YouTube videos in our dataset that were recorded using handheld cameras without stabilizers. Consequently, our model occasionally generates videos with noticeable shakiness. We illustrate one stable and one shaky example in Fig. 13. While the presence of shakiness doesn't necessarily constitute a failure – as it can contribute to a sense of realism – it may not consistently align with user preferences. To address this concern, we can simply create additional examples and enable users to pick their preferred style.

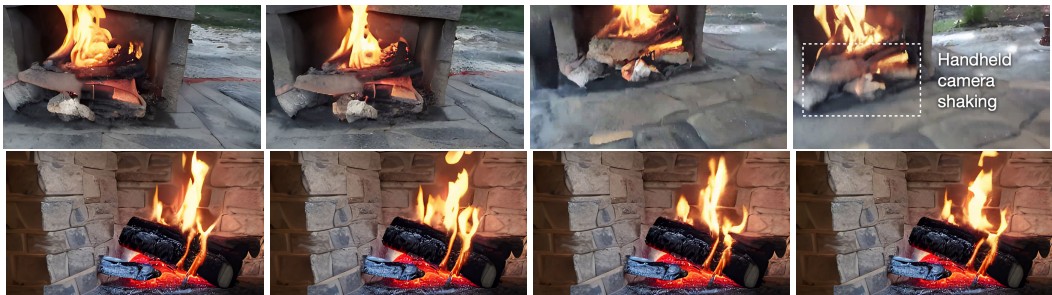

"A fire roars in a fireplace"

Figure 13: Cases of occasional handheld camera shaking.

## E.4 MORE SAMPLES

We demonstrate larger figures and more generated samples of our method in Figs. 14-18. The generated results exhibit high quality, characterized by clear motion, abundant detail, and accurate semantic alignment.

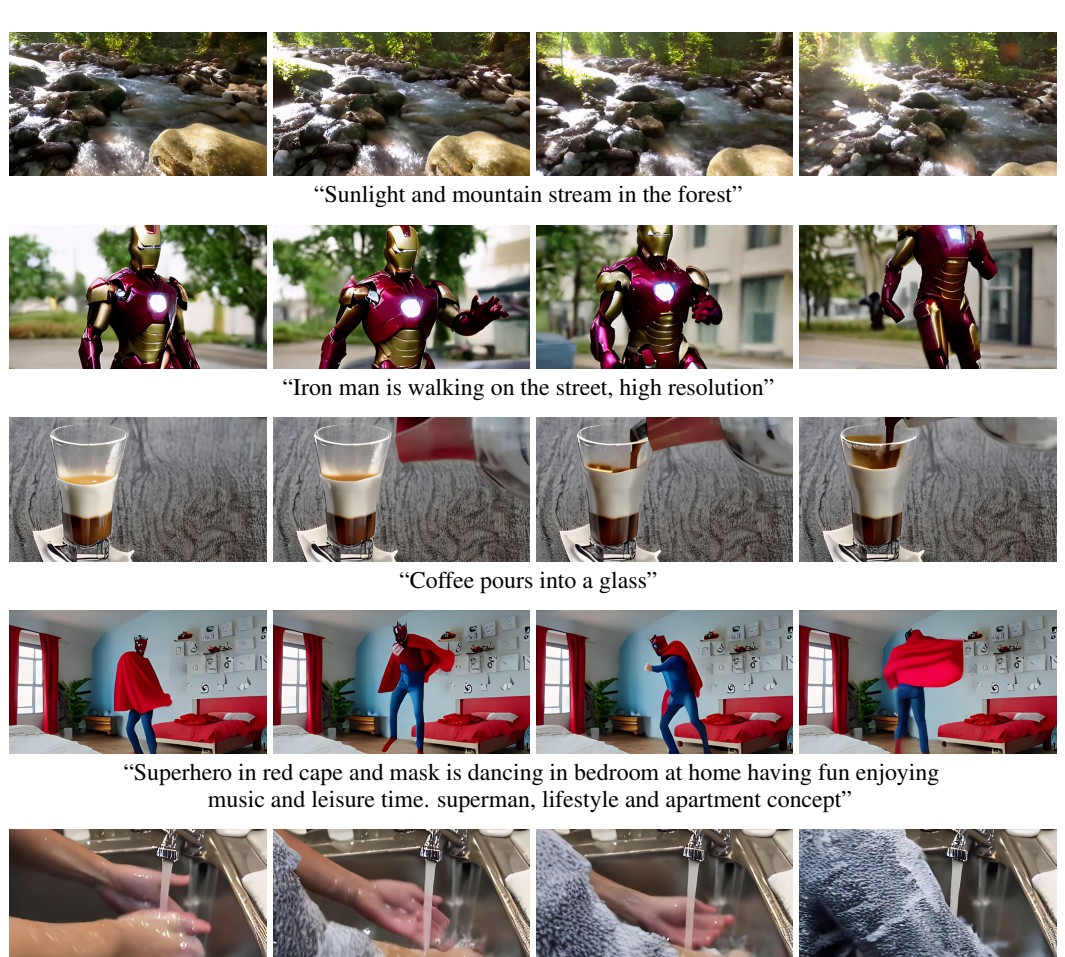

"Sunlight and mountain stream in the forest"

"Iron man is walking on the street, high resolution"

"Coffee pours into a glass"

"Superhero in red cape and mask is dancing in bedroom at home having fun enjoying music and leisure time. superman, lifestyle and apartment concept"

"Washing hands with water"

Figure 14: More generated samples of our VideoFactory. We can observe high-quality generated results with clear motion, rich detail, and well semantic alignment. Video resolution: 1376×768.

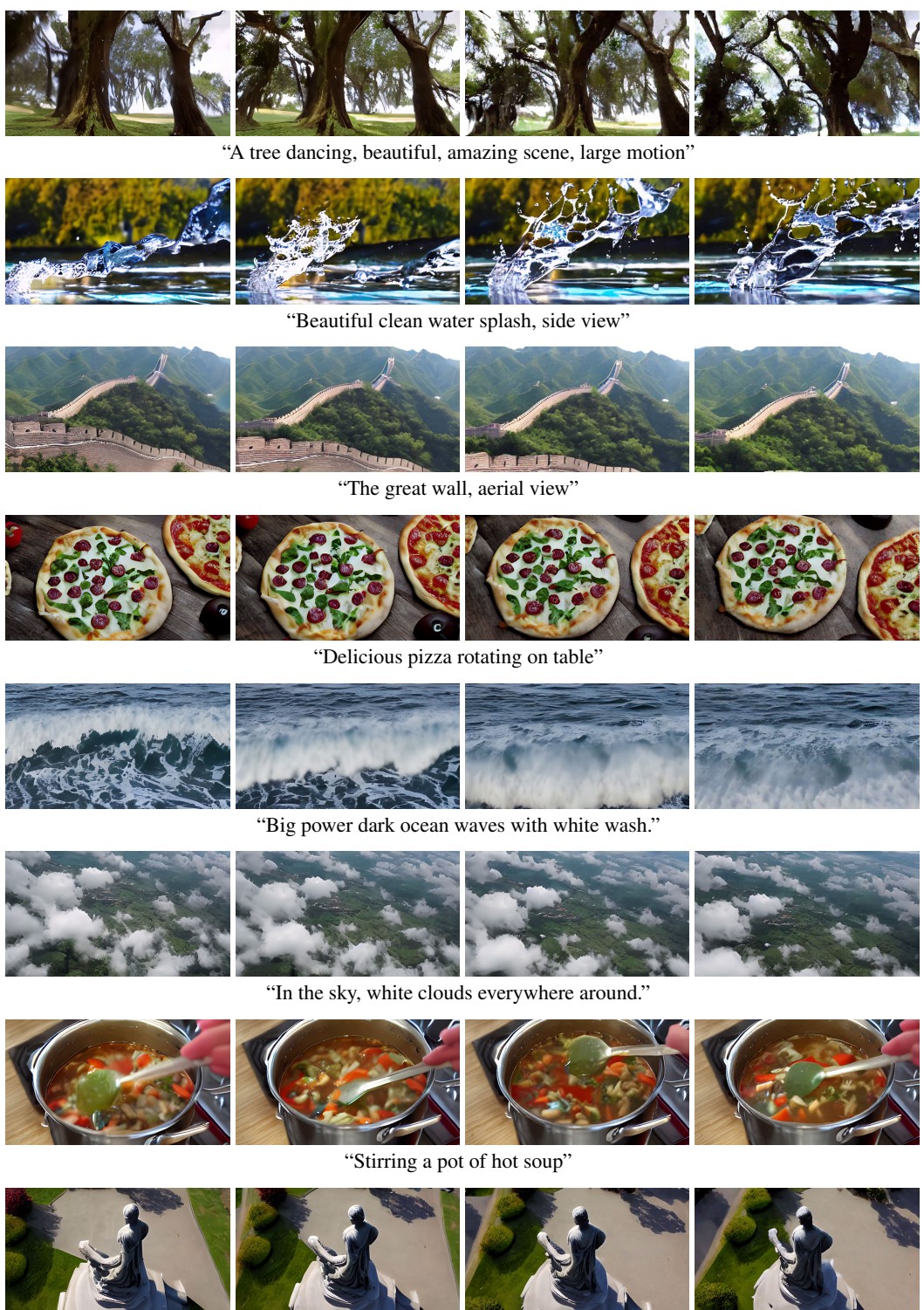

"A tree dancing, beautiful, amazing scene, large motion"

"Beautiful clean water splash, side view"

"The great wall, aerial view"

"Delicious pizza rotating on table"

"Big power dark ocean waves with white wash."

"In the sky, white clouds everywhere around."

"Stirring a pot of hot soup"

"Aerial view of a statue, rotation loop. beautiful light, high contrast"

Figure 15: More generated samples of our VideoFactory. We can observe high-quality generated results with clear motion, rich detail, and well semantic alignment. Video resolution: 1376×768.

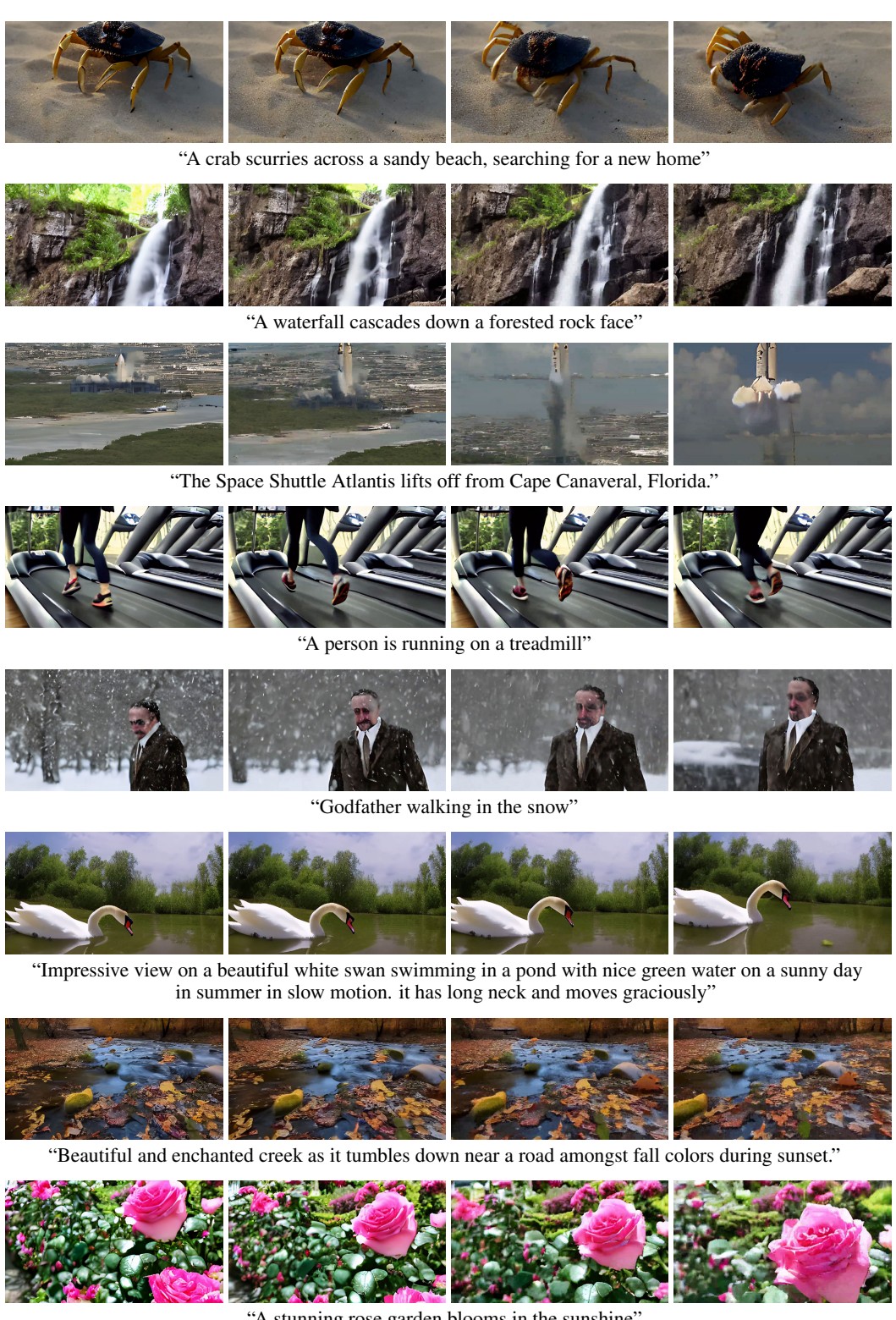

"A crab scurries across a sandy beach, searching for a new home"

"A waterfall cascades down a forested rock face"

"The Space Shuttle Atlantis lifts off from Cape Canaveral, Florida."

"A person is running on a treadmill"

"Godfather walking in the snow"

"Impressive view on a beautiful white swan swimming in a pond with nice green water on a sunny day in summer in slow motion. it has long neck and moves graciously"

"Beautiful and enchanted creek as it tumbles down near a road amongst fall colors during sunset."

"A stunning rose garden blooms in the sunshine"

Figure 16: More generated samples of our VideoFactory. We can observe high-quality generated results with clear motion, rich detail, and well semantic alignment. Video resolution: 1376×768.

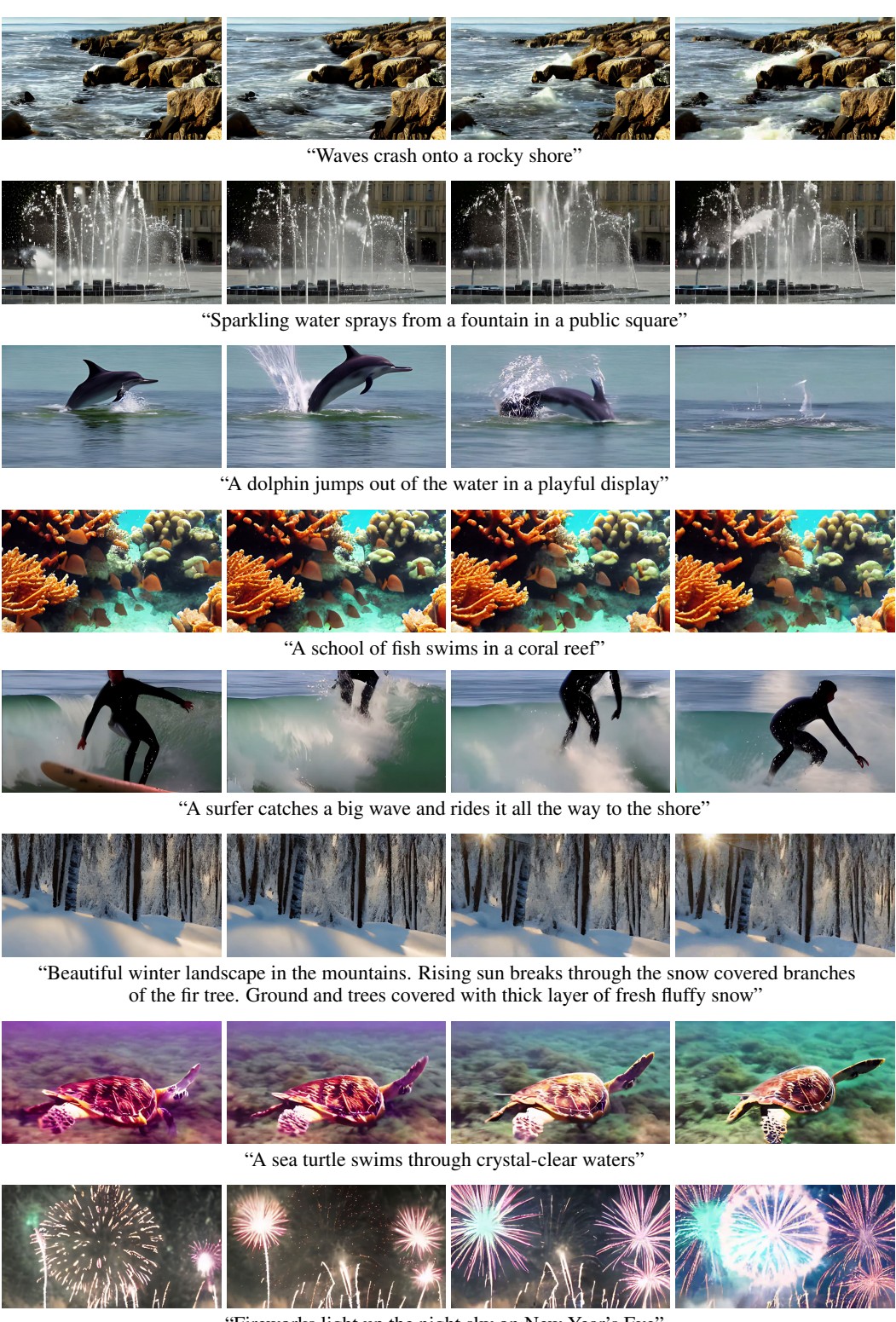

"Waves crash onto a rocky shore"

"Sparkling water sprays from a fountain in a public square"

"A dolphin jumps out of the water in a playful display"

"A school of fish swims in a coral reef"

"A surfer catches a big wave and rides it all the way to the shore"

"Beautiful winter landscape in the mountains. Rising sun breaks through the snow covered branches of the fir tree. Ground and trees covered with thick layer of fresh fluffy snow"

"A sea turtle swims through crystal-clear waters"

"Fireworks light up the night sky on New Year's Eve"

Figure 17: More generated samples of our VideoFactory. We can observe high-quality generated results with clear motion, rich detail, and well semantic alignment. Video resolution: 1376×768.

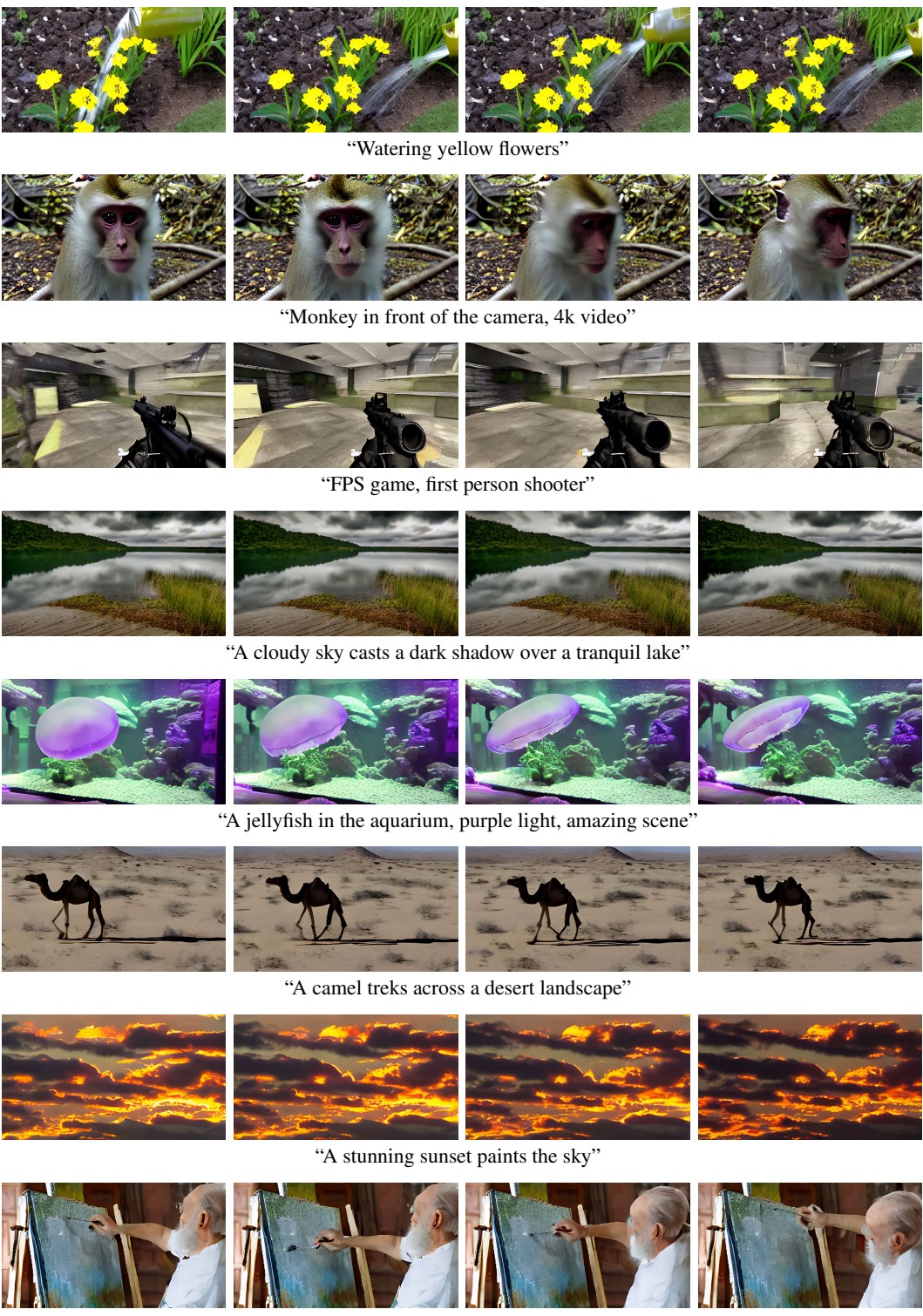

"Watering yellow flowers"

"Monkey in front of the camera, 4k video"

"FPS game, first person shooter"

"A cloudy sky casts a dark shadow over a tranquil lake"

"A jellyfish in the aquarium, purple light, amazing scene"

"A camel treks across a desert landscape"

"A stunning sunset paints the sky"

"An old painter is painting in the room"

Figure 18: More generated samples of our VideoFactory. We can observe high-quality generated results with clear motion, rich detail, and well semantic alignment. Video resolution: 1376×768.

### E.5   SAMPLES ON WEBVID

Alongside the qualitative experimental results shown in the main paper, here we present generated samples on captions sourced from the WebVid dataset in Fig. 19.

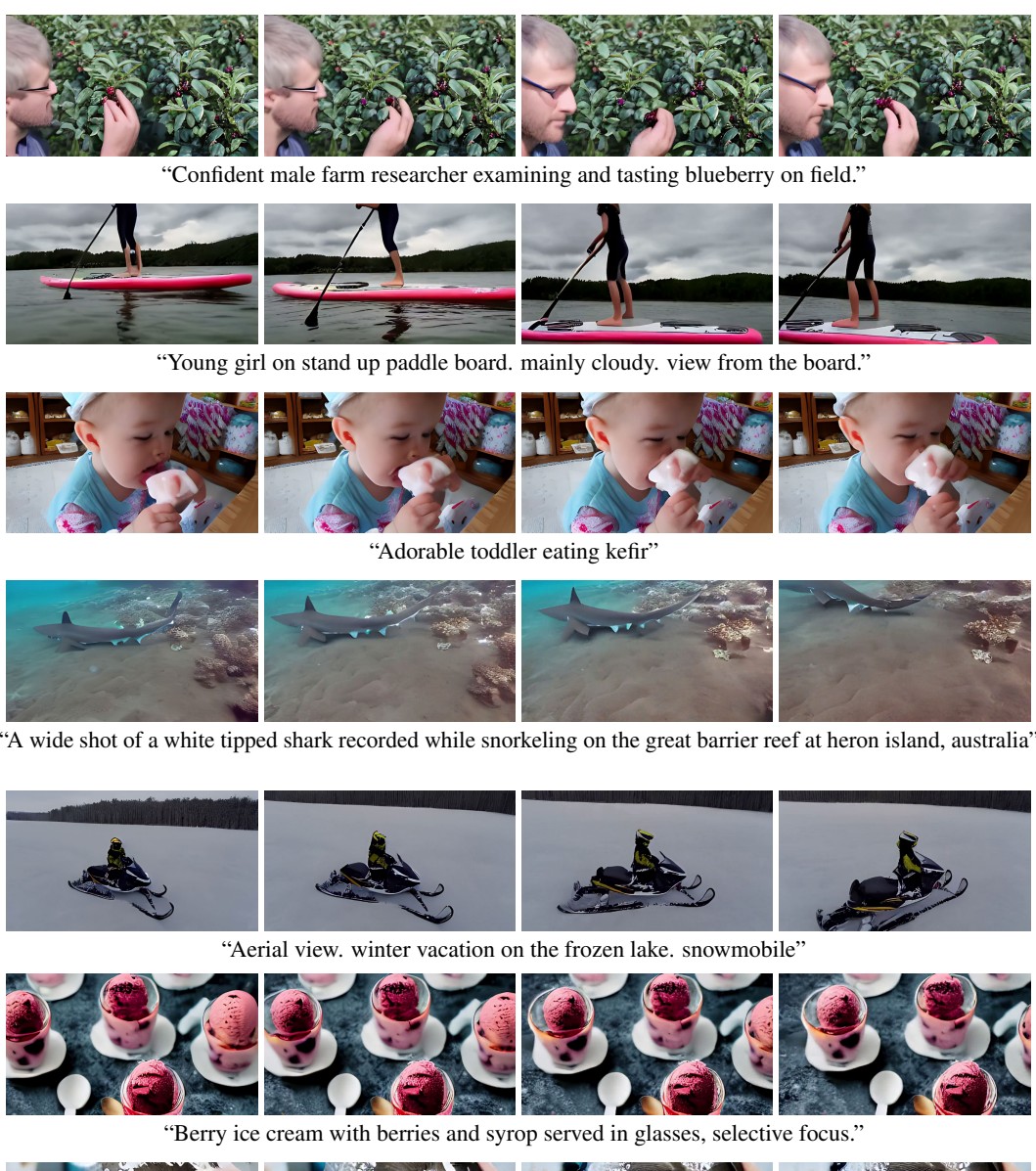

"Confident male farm researcher examining and tasting blueberry on field."

"Young girl on stand up paddle board. mainly cloudy. view from the board."

"Adorable toddler eating kefir"

"A wide shot of a white tipped shark recorded while snorkeling on the great barrier reef at heron island, australia"

"Aerial view. winter vacation on the frozen lake. snowmobile"

"Berry ice cream with berries and syrop served in glasses, selective focus."

"Hair stylist combs the hair of the bride to be"

Figure 19: Generated samples of our VideoFactory on WebVid. Video resolution: 1376×768.

### E.6 SAMPLES ON MSRVTT

Besides the qualitative experimental results showcased in the main paper, we also offer generated samples from the MSRVTT dataset in Fig. 20 for further examination.

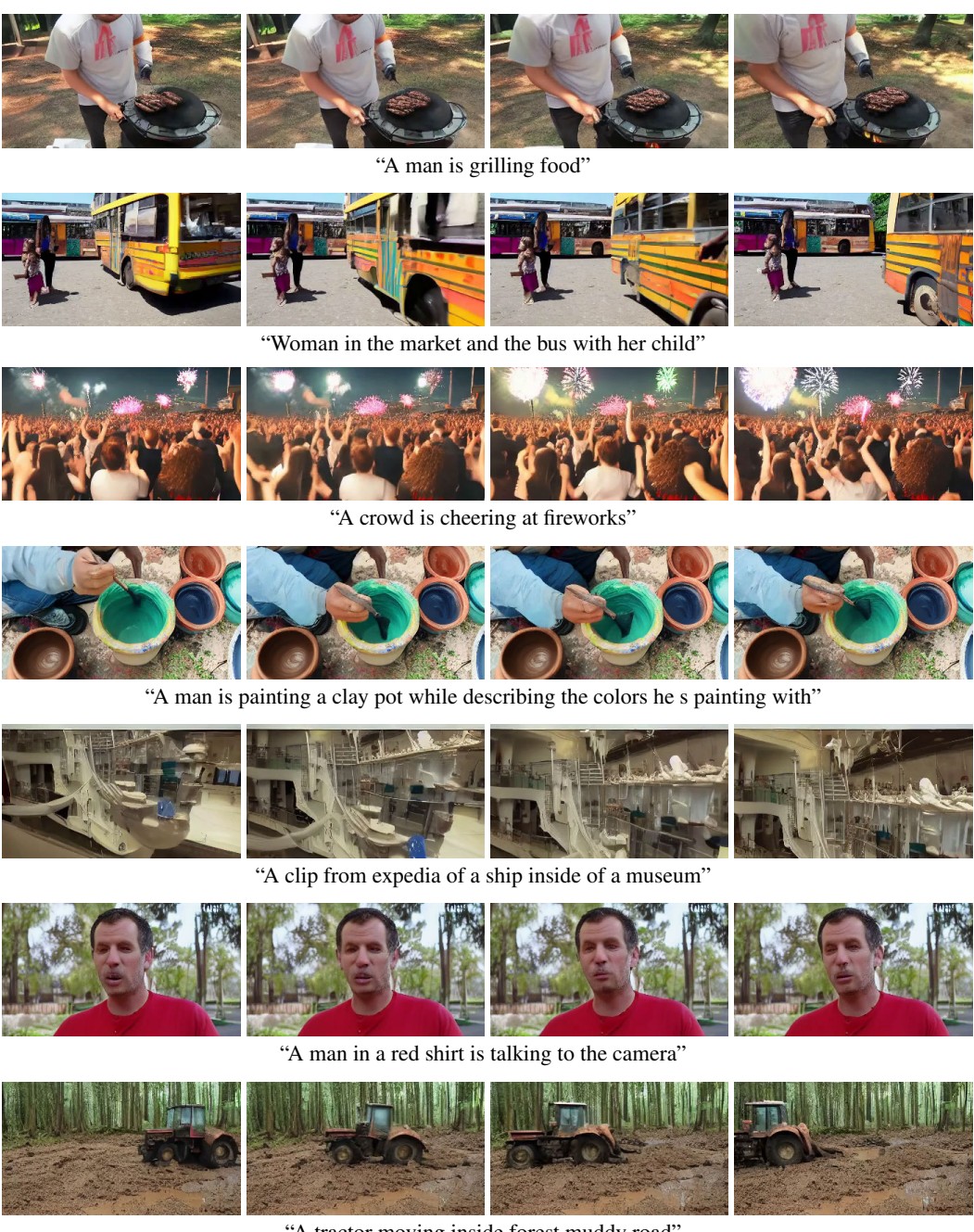

"A man is grilling food"

"Woman in the market and the bus with her child"

"A crowd is cheering at fireworks"

"A man is painting a clay pot while describing the colors he s painting with"

"A clip from expedia of a ship inside of a museum"

"A man in a red shirt is talking to the camera"

"A tractor moving inside forest muddy road"

Figure 20: Generated samples of our VideoFactory on MSR-VTT. Note that we do not use super-resolution on MSR-VTT for simplicity. Resolution: 344×192

### E.7 SAMPLES ON UCF101

In addition to the qualitative experimental results presented in the main paper, we also provide generated samples on the UCF101 dataset in Fig. 21.

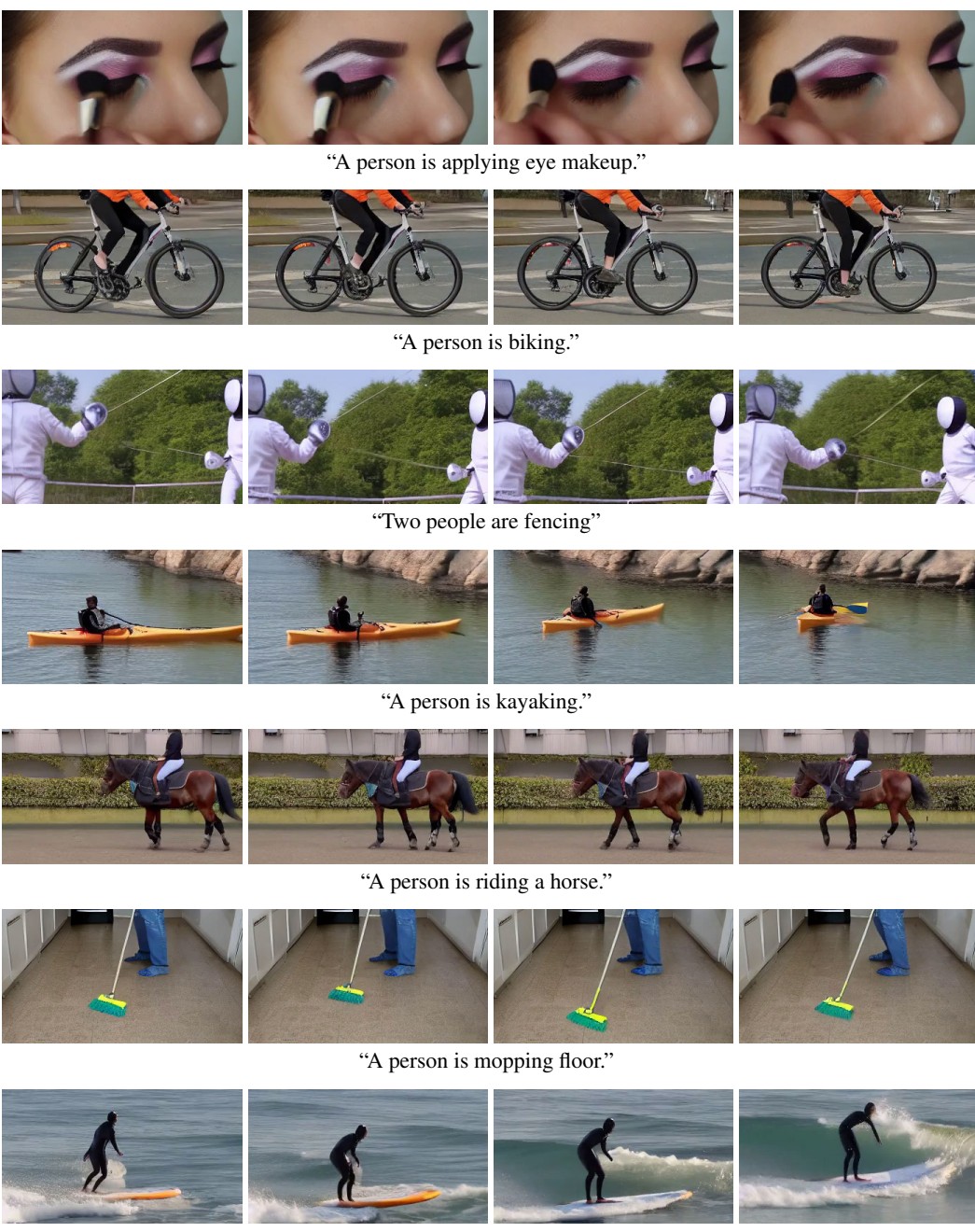

Figure 21: Generated samples of our VideoFactory on UCF101. Note that we do not use super-resolution on UCF101 for simplicity. Resolution: 344×192

