# OpenReview forum: "VideoFactory: Swap Attention in Spatiotemporal Diffusions for Text-to-Video Generation"
_ICLR.cc/2024/Conference — ICLR 2024 Conference Withdrawn Submission_

### Official Review · Reviewer_qRMz · 2023-10-28

**Soundness:** 2 fair
**Presentation:** 2 fair
**Contribution:** 2 fair
**Rating:** 5
**Confidence:** 4

**Summary:**

The paper presents a research work on text-to-video generation using a video diffusion model. The primary contribution of the paper lies in the exploration and proposal of a "swapped attention" structure for the temporal attention component within the diffusion model. Moreover, the authors introduce a comprehensive dataset comprising 130 million text-video pairs, featuring captions and resolutions of 720p or higher. Through experiments, the authors demonstrate that their proposed method achieves comparable or superior performance when compared to recent approaches such as Imagen Video and VideoLDM.

**Strengths:**

1. This paper conducts an investigation into the attention component of the U-Net model, specifically focusing on the process of fusing temporal and spatial features.
2. The paper introduces a high-quality dataset that is particularly noteworthy. The dataset includes fine-grained caption annotations and exhibits a high level of image quality (which is also evident in the generated results). Importantly, the authors plan to make this dataset publicly available, which will significantly contribute to the advancement of the research community.

**Weaknesses:**

The paper falls short in terms of introducing substantial technical innovation since there have been extensive discussions on the fusion of temporal features in the field of video classification. Several existing papers, such as ViViT, Timesformer, Pseudo3D, and CSN, have already explored and expanded upon this topic, providing potential avenues for further development and adaptation in the context of the Video Diffusion Model. The proposed "swapped attention" mechanism in this paper does not appear to significantly impact the task of video generation itself.

[1] ViViT: A Video Vision Transformer, https://arxiv.org/abs/2103.15691
[2] Timesformer, https://arxiv.org/abs/2102.05095
[3] Learning Spatio-Temporal Representation with Pseudo-3D Residual Networks, https://arxiv.org/abs/1711.10305

**Questions:**

Has the author investigated more forms of temporal-spatial feature fusion, and what are their effects in the Video Generation? Even among recent methods related to video diffusion, there have been numerous discussions on attention mechanisms, such as the work mentioned in https://arxiv.org/abs/2212.11565. Can the authors provide comparisons with these existing approaches?

---

### Official Review · Reviewer_Nqrs · 2023-10-29

**Soundness:** 2 fair
**Presentation:** 3 good
**Contribution:** 2 fair
**Rating:** 5
**Confidence:** 5

**Summary:**

This paper proposes a new attention mechanism, swapped cross-attention, to better fuse the spatial and temporal information in the video diffusion models. They also collected a text-video dataset called HD-VG-130M which contains YouTube videos and BLIP-2-generated captions.

**Strengths:**

The idea of improving spatial and temporal information exchange in the model design for video synthesis while preserving efficiency is relatively novel and useful. The paper is easy to follow, and the presentation is clear. The ablation experiments on the proposed attention are useful in understanding the proposed.

**Weaknesses:**

I'm not fully convinced by the usefulness of the collected dataset. The paper uses the caption of a keyframe in the video generated by a BLIP-2 model. However, the caption model does not have the knowledge of the temporal information of the video. This can be an issue, given my previous experience with running BLIP-2 on the WebVid-10M videos. For example, given a video of a group of people dancing, the model generated the caption "a group of people standing next to each other." Also, from the statistics of Figure 2, most of the captions have 7 - 11 words and are relatively short. Based on my experience of training text-to-video models, the quality of the captions is very important.

My other major concern is about the quantitative evaluation. The paper does not report the numbers for all the valid baselines and metrics. For example, Make-A-Video achieves FVD=367.23 on the zero-shot UCF evaluation. In addition, previous studies also report IS on the UCF dataset and FID-vid on the MSR-VTT dataset.

Although many ablation experiments are performed, I still recommend comparing the proposed attention with the 2D + 1D attention that was used in many previous studies. I might have missed it since I'm confused by Tables 2 and 3. They seem to both compare different attention mechanisms, but why is memory and computation analysis only done on the attention mechanism in Table 2?

The model underperforms Imagen Video in human study.

It seems that no code and model are planned to be released.

**Questions:**

1. What is the model size trained in Tables 2 and 3? The improvement of FVD could be more interpretable if they ablations are done on the standard benchmarks like the UCF-101 dataset.

2. What is the inference time of the videos of different resolutions?

3. In the last line of Equation (2), should it be z^{l+1}?

**Details Of Ethics Concerns:**

This paper contains a video dataset collected from YouTube, which may need ethics review.

---

### Official Review · Reviewer_jziR · 2023-10-31

**Soundness:** 3 good
**Presentation:** 3 good
**Contribution:** 3 good
**Rating:** 6
**Confidence:** 3

**Summary:**

The paper proposes an architectural modification of the UNet architecture used in video diffusion model. The modification, named Swapped spatiotemporal Cross-Attention, is based on insight and observations on how various forms of attention affect diffusion models' spatial and temporal consistency, and is motivated experimentally, achieving a strong balance between computational cost and generation quality. Furthermore, the paper also proposes a novel dataset collected and designed specifically for generative video modelling. Models trained on that dataset further improve in generation quality.

**Strengths:**

* Well-written paper centred around the idea of modifying self-attention in the UNet architecture used for video generation with diffusion models.
* Insights into how suitable various datasets are for video generation (e.g. Table 1 and Section 3) and how various architectural choices affect video generation metrics (Section 5.2 and Table 2) are very interesting. And more generally, these insights may extend beyond generative modelling (e.g. web-scale pre-training or supervised learning on video).
* The proposed modifications are supported by empirical results, achieving strong improvements prior models, and ablations that show a clear benefit of using the proposed architecture.

**Weaknesses:**

* My biggest set of questions centres around the proposed dataset and its creation process. It appears that the videos were sourced from YouTube. How were the videos chosen? under which licenses were the original videos uploaded? Under which license is the resulting dataset available? How will the changes in the original video's licenses be reflected in the dataset? What steps were taken to mitigate any sampling bias in the data? In general, I felt that dataset creation aspect of the paper was not adequately covered, and deserves more attention.

**Questions:**

* Table 7 is somewhat unclear. The column "Param Ratio" does not seem to really belong there, as the other columns as far as I could tell, are scores. The caption mentions that the scores are are percentages, but I guess they are fractions? Please specify whether the results of these AB tests are statistically significant.
* In section Section 5.3 in several evals (e.g. MSR-VTT and Web-Vid-10M) the validation sets are sampled randomly. In the case of WebVid-10M the sampled validation set is also rather small (5K samples). How robust are the results w.r.t sampling multiple times?
* Will the HD-VG-130M dataset be made easily available to be used by other researchers?

**Details Of Ethics Concerns:**

The paper proposes a new dataset sourced from YouTube. It's not clear from the description if and how original video licences were respected.

---

### Official Review · Reviewer_NUsL · 2023-10-31

**Soundness:** 2 fair
**Presentation:** 3 good
**Contribution:** 2 fair
**Rating:** 5
**Confidence:** 4

**Summary:**

This paper presents a text-to-video generation model and a large-scale dataset of text/video pairs. The video generator is initialized from ModelScope (which uses the LDM backbone under the hood) and fine-tines on the collected dataset with a novel "swapped spatio-temporal attention" module. Then, a GAN-based super-resolution model is trained on top of it. For the dataset, the authors collect HD youtube videos, split them into clips with PySceneDetect and annotate the clips with BLIP-2 using the middle frame. For the video generator, it is then conducts a rigorous evaluation on UCF101, MSR-VTT, WebVid-10M and also includes human evaluation results.

**Strengths:**

- The model is quite cheap to train, though it fine-tunes from ModelScope (VideoFusion), which fine-tunes from LDM.
- The proposed HD-VG-130M dataset is a solid contribution and should help the community develop better generators.
- The evaluation is quite thorough and encompasses text-video alignment, visual quality and the human studies. The paper also makes a good effort comparing to closed-source models (Make-a-Video and ImagenVideo). And also includes the failure case study.
- I appreciate that training costs have been reported, and also throughpout/memory requirements. I believe that reporting such information to be crucial for foundational DL models.
- The paper is well written and provides rich information about both the dataset, the main model, and the training information.

**Weaknesses:**

- When comparing FVDs on UCF-101, the paper included many previous works — but only if their FVD score is higher: https://paperswithcode.com/sota/video-generation-on-ucf-101. In this way, it omitted Make-A-Video, VDIM, LVDM, etc. despite comparing it in other regards. I do not see how it could be justified.
- The contributions of the proposed components (datasets, new attention mechanism, etc) are entangled between each other and the ablations are confusing. Table 2 shows the scores for different components on WebVid, but there is no information about the ablation experiments: have each model was trained anew from ModelScope? For how many steps? Was it trained on webvid or HD-VG + LAION? I do not intuitively understand how swapped spatio-temporal attention can outperform just the "brute-force" global attention, if the latter one is given enough compute/training data.
- The dataset annotation with captions is fairly simple and cannot capture any dynamics. For example, if a person comes to a table and takes a cup, then the annotation would be something like "a person walks towards a table", because only the middle frame was used for static caption annotation.

**Questions:**

- Do I get it right that the model takes ~2.5 days of training on 32 V100s? Also, am I understanding this correctly that in total the model has been trained for ~40k steps with the batch size of 128, which implies that it has seen only ~5M videos (+ images)?
- Why does Table 4 omit the methods which have better FVDs? (https://paperswithcode.com/sota/video-generation-on-ucf-101)
- I suspect problems in human studies. I do not understand how Imagen-Video or Make-a-Video can be so much worse in terms of the performance in Table 7. Could you please share the videos which were used for the human studies?
- For ablations, is it possible to train a model completely from scratch (without initializing it from anything) on some small-scale dataset, like UCF-101?

---

### Author Response · Authors · 2023-11-21
**Thank you!**

We're sincerely grateful to all the reviewers for their careful review and valuable comments. After considering all the suggestions and concerns, we plan to withdraw the current manuscript and improve it in the future version.